# DEEP NETWORKS FROM THE PRINCIPLE OF RATE REDUCTION

## ABSTRACT

This work attempts to interpret modern deep (convolutional) networks from the principles of rate reduction and (shift) invariant classification. We show that the basic iterative gradient ascent scheme for maximizing the rate reduction of learned features naturally leads to a deep network, one iteration per layer. The architectures, operators (linear or nonlinear), and parameters of the network are all explicitly constructed layer-by-layer in a forward propagation fashion. All components of this "white box" network have precise optimization, statistical, and geometric interpretation. Our preliminary experiments indicate that such a network can already learn a good discriminative deep representation without any back propagation training. Moreover, all linear operators of the so-derived network naturally become multi-channel convolutions when we enforce classification to be rigorously shift-invariant. The derivation also indicates that such a convolutional network is significantly more efficient to learn and construct in the spectral domain.

## 1 INTRODUCTION AND MOTIVATION

In recent years, various deep (convolution) network architectures such as AlexNet (Krizhevsky et al., 2012), VGG (Simonyan & Zisserman, 2015), ResNet (He et al., 2016), DenseNet (Huang et al., 2017), Recurrent CNN, LSTM (Hochreiter & Schmidhuber, 1997), Capsule Networks (Hinton et al., 2011), etc., have demonstrated very good performance in classification tasks of real-world datasets such as speeches or images. Nevertheless, almost all such networks are developed through years of empirical *trial and error*, including both their architectures/operators and the ways they are to be effectively trained. Some recent practices even take to the extreme by searching for effective network structures and training strategies through extensive random search techniques, such as Neural Architecture Search (Zoph & Le, 2017; Baker et al., 2017), AutoML (Hutter et al., 2019), and Learning to Learn (Andrychowicz et al., 2016).

Despite tremendous empirical advances, there is still a lack of rigorous theoretical justification of the need or reasons for "deep" network architectures and a lack of fundamental understanding of the associated operators (e.g. multi-channel convolution and nonlinear activation) in each layer. As a result, deep networks are often designed and trained heuristically and then used as a "black box." There have been a severe lack of guiding principles for each of the stages: For a given task, how wide or deep the network should be? What are the roles and relationships among the multiple (convolution) channels? Which parts of the networks need to be learned and trained and which can be determined in advance? How to evaluate the optimality of the resulting network? As a consequence, besides empirical evaluation, it is usually impossible to offer any rigorous guarantees for certain performance of a trained network, such as invariance to transformation (Azulay & Weiss, 2018; Engstrom et al., 2017) or overfitting noisy or even arbitrary labels (Zhang et al., 2017).

In this paper, we do not intend to address all these questions but we would attempt to offer a plausible interpretation of deep (convolution) neural networks by deriving a class of deep networks *from first principles*. We contend that all key features and structures of modern deep (convolution) neural networks can be naturally derived from optimizing a principled objective, namely the *rate reduction* recently proposed by Yu et al. (2020), that seeks a compact discriminative (invariant) representation of the data. More specifically, the basic iterative *gradient ascent* scheme for optimizing the objective naturally takes the form of a deep neural network, one layer per iteration.

This principled approach brings a couple of nice surprises: First, architectures, operators, and parameters of the network can be constructed explicitly layer-by-layer in a *forward propagation* fashion, and all inherit precise optimization, statistical and geometric interpretation. As result, the so constructed "white box" deep network already gives a good discriminative representation (and achieves good classification performance) *without any back propagation* for training the deep network. Second, in the case of seeking a representation *rigorously* invariant to shift or translation, the network naturally lends itself to a multi-channel convolutional network. Moreover, the derivation indicates such a convolutional network is computationally more efficient to learn and construct in the *spectral*

*(Fourier) domain*, analogous to how neurons in the visual cortex encode and transit information with their spiking frequencies (Eliasmith & Anderson, 2003; Belitski et al., 2008).

## 2 TECHNICAL APPROACH

Consider a basic classification task: given a set of $m$ samples $\boldsymbol{X} \doteq [\boldsymbol{x}^1, \ldots, \boldsymbol{x}^m] \in \mathbb{R}^{n \times m}$ and their associated memberships $\boldsymbol{\pi}(\boldsymbol{x}^i) \in [k]$ in $k$ different classes, a deep network is typically used to model a direct mapping from the input data $\boldsymbol{x} \in \mathbb{R}^n$ to its class label $f(\boldsymbol{x}, \boldsymbol{\theta}) : \boldsymbol{x} \mapsto \boldsymbol{y} \in \mathbb{R}^k$, where $\boldsymbol{y}$ is typically a "one-hot" vector encoding the membership information $\boldsymbol{\pi}(\boldsymbol{x})$: the $j$-th entry of $\boldsymbol{y}$ is 1 iff $\boldsymbol{\pi}(\boldsymbol{x}) = j$. The parameters $\boldsymbol{\theta}$ of the network is typically learned to minimize certain prediction loss, say the cross entropy loss, via gradient-descent type back propagation. Although this popular approach provides people a direct and effective way to train a network that predicts the class information, the so learned representation is however implicit and lacks clear interpretation.

### 2.1 PRINCIPLE OF RATE REDUCTION AND GROUP INVARIANCE

**The Principle of Maximal Coding Rate Reduction.** To help better understand features learned in a deep network, the recent work of Yu et al. (2020) has argued that the goal of (deep) learning is to learn a compact discriminative and diverse feature representation[1] $\boldsymbol{z} = f(\boldsymbol{x}) \in \mathbb{R}^n$ of the data $\boldsymbol{x}$ before any subsequent tasks such as classification: $\boldsymbol{x} \xrightarrow{f(\boldsymbol{x})} \boldsymbol{z} \xrightarrow{h(\boldsymbol{z})} \boldsymbol{y}$. To be more precise, instead of directly fitting the class label $\boldsymbol{y}$, a principled objective is to learn a feature map $f(\boldsymbol{x}) : \boldsymbol{x} \mapsto \boldsymbol{z}$ which transforms the data $\boldsymbol{x}$ onto a set of most discriminative low-dimensional linear subspaces $\{\mathcal{S}^j\}_{j=1}^k \subset \mathbb{R}^n$, one subspace $\mathcal{S}^j$ per class $j \in [k]$.

Let $\boldsymbol{Z} \doteq [\boldsymbol{z}^1, \ldots, \boldsymbol{z}^m] = [f(\boldsymbol{x}^1), \ldots, f(\boldsymbol{x}^m)]$ be the features of the given samples $\boldsymbol{X}$. WLOG, we may assume all features $\boldsymbol{z}^i$ are normalized to be of unit norm: $\boldsymbol{z}^i \in \mathbb{S}^{n-1}$. For convenience, let $\boldsymbol{\Pi}^j \in \mathbb{R}^{m \times m}$ be a diagonal matrix whose diagonal entries encode the membership of samples/features belong to the $j$-th class: $\boldsymbol{\Pi}^j(i, i) = \boldsymbol{\pi}(\boldsymbol{x}^i) = \boldsymbol{\pi}(\boldsymbol{z}^i)$. Then based on principles from lossy data compression (Ma et al., 2007), Yu et al. (2020) has suggested that the optimal representation $\boldsymbol{Z}_\star \subset \mathbb{S}^{n-1}$ should maximize the following coding rate reduction objective, known as the MCR$^2$ principle:

$$\textit{Rate Reduction:} \quad \Delta R(\boldsymbol{Z}) \doteq \underbrace{\frac{1}{2} \log \det \left( \boldsymbol{I} + \alpha \boldsymbol{Z} \boldsymbol{Z}^* \right)}_{R(\boldsymbol{Z})} - \underbrace{\sum_{j=1}^k \frac{\gamma_j}{2} \log \det \left( \boldsymbol{I} + \alpha_j \boldsymbol{Z} \boldsymbol{\Pi}^j \boldsymbol{Z}^* \right)}_{R_c(\boldsymbol{Z}, \boldsymbol{\Pi})}, \quad (1)$$

where $\alpha = n/(m\epsilon^2)$, $\alpha_j = n/(\text{tr}(\boldsymbol{\Pi}^j)\epsilon^2)$, $\gamma_j = \text{tr}(\boldsymbol{\Pi}^j)/m$ for $j = 1, \ldots, k$. Given a prescribed quantization error $\epsilon$, the first term $R$ of $\Delta R(\boldsymbol{Z})$ measures the total coding length for all the features $\boldsymbol{Z}$ and the second term $R_c$ is the sum of coding lengths for features in each of the $k$ classes.

In Yu et al. (2020), the authors have shown the optimal representation $\boldsymbol{Z}_\star$ that maximizes the above object indeed has desirable nice properties. Nevertheless, they adopted a conventional deep network (e.g. the ResNet) as a black box to model and parameterize the feature mapping: $\boldsymbol{z} = f(\boldsymbol{x}, \boldsymbol{\theta})$. It has empirically shown that with such a choice, one can effectively optimize the MCR$^2$ objective and obtain discriminative and diverse representations for classifying real image data sets.

However, there remain several unanswered problems. Although the resulting feature representation is more interpretable, the network itself is still not. It is *not* clear why any chosen network is able to optimize the desired MCR$^2$ objective: Would there be any potential limitations? The good empirical results (say with a ResNet) do not necessarily justify the particular choice in architectures and operators of the network: Why is a layered model necessary, how wide and deep is adequate, and is there any rigorous justification for the use of convolutions and nonlinear operators used? In Section 2.2, we show that using gradient ascent to maximize the rate reduction $\Delta R(\boldsymbol{Z})$ naturally leads to a "white box" deep network that represents such a mapping. All linear/nonlinear operators and parameters of the network are *explicitly constructed in a purely forward propagation fashion*.

**Group Invariant Rate Reduction.** So far, we have considered the data and features as vectors. In many applications, such as serial data or imagery data, the semantic meaning (labels) of the data and their features are *invariant* to certain transformations $\mathfrak{g} \in \mathbb{G}$ (for some group $\mathbb{G}$) (Cohen & Welling, 2016). For example, the meaning of an audio signal is invariant to shift in time; and the identity of an object in an image is invariant to translation in the image plane. Hence, we prefer the feature mapping $f(\boldsymbol{x}, \boldsymbol{\theta})$ is rigorously invariant to such transformations:

---

[1] To simplify the presentation, we assume for now that the feature $\boldsymbol{z}$ and $\boldsymbol{x}$ have the same dimension $n$. But in general they can be different as we will soon see, say in the case $\boldsymbol{z}$ is multi-channel extracted from $\boldsymbol{x}$.

$$\text{Group Invariance:} \quad f(\boldsymbol{x} \circ \mathfrak{g}, \boldsymbol{\theta}) \sim f(\boldsymbol{x}, \boldsymbol{\theta}), \quad \forall \mathfrak{g} \in \mathbb{G}, \tag{2}$$

where "$\sim$" indicates two features belonging to the same equivalent class. The recent work of Zaheer et al. (2017); Maron et al. (2020) characterize properties of networks and operators for set permutation groups. Nevertheless, it remains challenging to learn features via a deep network that are *guaranteed* to be invariant even to simple transformations such as translation and rotation (Azulay & Weiss, 2018; Engstrom et al., 2017). In Section 2.3, we show that the MCR$^2$ principle is compatible with invariance in a very natural and precise way: we only need to assign all transformed versions $\{\boldsymbol{x} \circ \mathfrak{g} \mid \mathfrak{g} \in \mathbb{G}\}$ into the same class as $\boldsymbol{x}$ and map them all to the same subspace $\mathcal{S}$.[2] We will rigorously show (in the Appendices) that, when the group $\mathbb{G}$ is (discrete) circular 1D shifting or 2D translation, the resulting deep network naturally becomes a *multi-channel convolution network*!

## 2.2 DEEP NETWORKS FROM MAXIMIZING RATE REDUCTION

**Gradient Ascent for Rate Reduction on the Training Samples.** First let us directly try to maximize the objective $\Delta R(\boldsymbol{Z})$ as a function in the training samples $\boldsymbol{Z} \subset \mathbb{S}^{n-1}$. To this end, we may adopt a (projected) *gradient ascent* scheme, for some step size $\eta > 0$:

$$\boldsymbol{Z}_{\ell+1} \;\propto\; \boldsymbol{Z}_\ell + \eta \cdot \left.\frac{\partial \Delta R}{\partial \boldsymbol{Z}}\right|_{\boldsymbol{Z}_\ell} \quad \text{subject to} \quad \boldsymbol{Z}_{\ell+1} \subset \mathbb{S}^{n-1}. \tag{3}$$

This scheme can be interpreted as how one should incrementally adjust locations of the current features $\boldsymbol{Z}_\ell$ in order for the resulting $\boldsymbol{Z}_{\ell+1}$ to improve the rate reduction $\Delta R(\boldsymbol{Z})$. Simple calculation shows that the gradient $\frac{\partial \Delta R}{\partial \boldsymbol{Z}}$ entails evaluating the following derivatives of the terms in (1):

$$\frac{1}{2}\left.\frac{\partial \log\det(\boldsymbol{I} + \alpha \boldsymbol{Z}\boldsymbol{Z}^*)}{\partial \boldsymbol{Z}}\right|_{\boldsymbol{Z}_\ell} = \underbrace{\alpha(\boldsymbol{I} + \alpha \boldsymbol{Z}_\ell \boldsymbol{Z}_\ell^*)^{-1}}_{\boldsymbol{E}_\ell \,\in \mathbb{R}^{n\times n}} \boldsymbol{Z}_\ell \quad \in \mathbb{R}^{n\times m}, \tag{4}$$

$$\frac{1}{2}\left.\frac{\partial \left(\gamma_j \log\det(\boldsymbol{I} + \alpha_j \boldsymbol{Z}\boldsymbol{\Pi}^j \boldsymbol{Z}^*)\right)}{\partial \boldsymbol{Z}}\right|_{\boldsymbol{Z}_\ell} = \gamma_j \underbrace{\alpha_j(\boldsymbol{I} + \alpha_j \boldsymbol{Z}_\ell \boldsymbol{\Pi}^j \boldsymbol{Z}_\ell^*)^{-1}}_{\boldsymbol{C}_\ell^j \,\in \mathbb{R}^{n\times n}} \boldsymbol{Z}_\ell \boldsymbol{\Pi}^j \quad \in \mathbb{R}^{n\times m}. \tag{5}$$

Notice that in the above, the matrix $\boldsymbol{E}_\ell$ only depends on $\boldsymbol{Z}_\ell$ hence it aims to *expand* all the features to increase the overall coding rate; the matrix $\boldsymbol{C}_\ell^j$ depends on features from each class and aims *compress* them to reduce the coding rate of each class. See Remark 1 in Appendix A for the geometric and statistic meaning of $\boldsymbol{E}_\ell$ and $\boldsymbol{C}_\ell^j$. Then the complete gradient $\frac{\partial \Delta R}{\partial \boldsymbol{Z}}\big|_{\boldsymbol{Z}_\ell}$ is of the following form:

$$\left.\frac{\partial \Delta R}{\partial \boldsymbol{Z}}\right|_{\boldsymbol{Z}_\ell} = \underbrace{\boldsymbol{E}_\ell}_{\text{Expansion}} \boldsymbol{Z}_\ell - \sum_{j=1}^{k} \gamma_j \underbrace{\boldsymbol{C}_\ell^j}_{\text{Compression}} \boldsymbol{Z}_\ell \boldsymbol{\Pi}^j \quad \in \mathbb{R}^{n\times m}. \tag{6}$$

**Gradient-Guided Feature Map Increment.** Notice that in the above, the gradient ascent considers all the features $\boldsymbol{Z}_\ell = [\boldsymbol{z}_\ell^1, \ldots, \boldsymbol{z}_\ell^m]$ as free variables. The increment $\boldsymbol{Z}_{\ell+1} - \boldsymbol{Z}_\ell = \eta \frac{\partial \Delta R}{\partial \boldsymbol{Z}}\big|_{\boldsymbol{Z}_\ell}$ does not yet give a transform on the entire feature domain $\boldsymbol{z}_\ell \in \mathbb{R}^n$. Hence, in order to find the optimal $f(\boldsymbol{x}, \boldsymbol{\theta})$ explicitly, we may consider constructing a small increment transform $g(\cdot, \boldsymbol{\theta}_\ell)$ on the $\ell$-th layer feature $\boldsymbol{z}_\ell$ to emulate the above (projected) gradient scheme:

$$\boldsymbol{z}_{\ell+1} \;\propto\; \boldsymbol{z}_\ell + \eta \cdot g(\boldsymbol{z}_\ell, \boldsymbol{\theta}_\ell) \quad \text{subject to} \quad \boldsymbol{z}_{\ell+1} \in \mathbb{S}^{n-1} \tag{7}$$

such that: $\left[g(\boldsymbol{z}_\ell^1, \boldsymbol{\theta}_\ell), \ldots, g(\boldsymbol{z}_\ell^m, \boldsymbol{\theta}_\ell)\right] \approx \frac{\partial \Delta R}{\partial \boldsymbol{Z}}\big|_{\boldsymbol{Z}_\ell}$. That is, we need to approximate the gradient flow $\frac{\partial \Delta R}{\partial \boldsymbol{Z}}$ that locally deforms each (training) feature $\{\boldsymbol{z}_\ell^i\}_{i=1}^m$ with a continuous mapping $g(\boldsymbol{z})$ defined on the entire feature space $\boldsymbol{z}_\ell \in \mathbb{R}^n$. See Remark 2 in Appendix A for conceptual connection and difference from the Neural ODE framework proposed by Chen et al. (2018).

By inspecting the structure of the gradient (6), it suggests that a natural candidate for the increment transform $g(\boldsymbol{z}_\ell, \boldsymbol{\theta}_\ell)$ is of the form:

$$g(\boldsymbol{z}_\ell, \boldsymbol{\theta}_\ell) \;\doteq\; \boldsymbol{E}_\ell \boldsymbol{z}_\ell - \sum_{j=1}^{k} \gamma_j \boldsymbol{C}_\ell^j \boldsymbol{z}_\ell \boldsymbol{\pi}^j(\boldsymbol{z}_\ell) \quad \in \mathbb{R}^n, \tag{8}$$

where $\boldsymbol{\pi}^j(\boldsymbol{z}_\ell) \in [0, 1]$ indicates the probability of $\boldsymbol{z}_\ell$ belonging to the $j$-th class.[3] Notice that the increment depends on 1). A linear map represented by $\boldsymbol{E}_\ell$ that depends only on statistics of all features from the preceding layer; 2). A set of linear maps $\{\boldsymbol{C}_\ell^j\}_{j=1}^k$ and memberships $\{\boldsymbol{\pi}^j(\boldsymbol{z}_\ell)\}_{j=1}^k$ of the features.

---

[2] Hence, any subsequent classifiers defined on the resulting set of subspaces will be automatically invariant to such transformations.

[3] Notice that on the training samples $\boldsymbol{Z}_\ell$, for which the memberships $\boldsymbol{\Pi}^j$ are known, the so defined $g(\boldsymbol{z}_\ell, \boldsymbol{\theta})$ gives exactly the values for the gradient $\frac{\partial \Delta R}{\partial \boldsymbol{Z}}\big|_{\boldsymbol{Z}_\ell}$.

Since we only have the membership $\boldsymbol{\pi}^j$ for the training samples, the function $g$ defined in (8) can only be evaluated on the training samples. To extrapolate the function $g$ to the entire feature space, we need to estimate $\boldsymbol{\pi}^j(\boldsymbol{z}_\ell)$ in its second term. In the conventional deep learning, this map is typically modeled as a deep network and learned from the training data, say via *back propagation*. Nevertheless, our goal here is not to learn a precise classifier $\boldsymbol{\pi}^j(\boldsymbol{z}_\ell)$ already. Instead, we only need a good enough estimate of the class information in order for $g$ to approximate the gradient $\frac{\partial \Delta R}{\partial \boldsymbol{Z}}$ well.

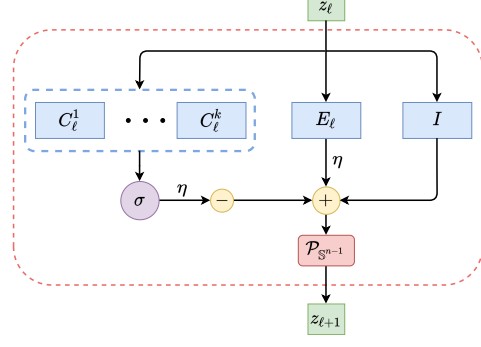

Figure 1: Layer structure of the **ReduNet**: from one iteration of gradient ascent for rate reduction.

From the geometric interpretation of the linear maps $\boldsymbol{E}_\ell$ and $\boldsymbol{C}_\ell^j$ given by Remark 1 in Appendix A, the term $\boldsymbol{p}_\ell^j \doteq \boldsymbol{C}_\ell^j \boldsymbol{z}_\ell$ can be viewed as projection of $\boldsymbol{z}_\ell$ onto the orthogonal complement of each class $j$. Therefore, $\|\boldsymbol{p}_\ell^j\|_2$ is small if $\boldsymbol{z}_\ell$ is in class $j$ and large otherwise. This motivates us to estimate its membership based on the following softmax function: $\widehat{\boldsymbol{\pi}}^j(\boldsymbol{z}_\ell) \doteq \frac{\exp\left(-\lambda\|\boldsymbol{C}_\ell^j \boldsymbol{z}_\ell\|\right)}{\sum_{j=1}^{k} \exp\left(-\lambda\|\boldsymbol{C}_\ell^j \boldsymbol{z}_\ell\|\right)} \in [0,1]$.

Hence the second term of (8) can be approximated by this estimated membership:[4]

$$\sum_{j=1}^{k} \gamma_j \boldsymbol{C}_\ell^j \boldsymbol{z}_\ell \boldsymbol{\pi}^j(\boldsymbol{z}_\ell) \approx \sum_{j=1}^{k} \gamma_j \boldsymbol{C}_\ell^j \boldsymbol{z}_\ell \cdot \widehat{\boldsymbol{\pi}}^j(\boldsymbol{z}_\ell) \;\; \doteq \;\; \boldsymbol{\sigma}\Big([\boldsymbol{C}_\ell^1 \boldsymbol{z}_\ell, \ldots, \boldsymbol{C}_\ell^k \boldsymbol{z}_\ell]\Big) \;\;\; \in \mathbb{R}^n, \qquad (9)$$

which is denoted as a nonlinear operator $\boldsymbol{\sigma}(\cdot)$ on outputs of the feature $\boldsymbol{z}_\ell$ through $k$ banks of filters: $[\boldsymbol{C}_\ell^1, \ldots, \boldsymbol{C}_\ell^k]$. Notice that the nonlinearality arises due to a "soft" assignment of class membership based on the feature responses from those filters. Overall, combining (7), (8), and (9), the increment feature transform from $\boldsymbol{z}_\ell$ to $\boldsymbol{z}_{\ell+1}$ now becomes:

$$\boldsymbol{z}_{\ell+1} \;\propto\; \boldsymbol{z}_\ell + \eta \cdot \boldsymbol{E}_\ell \boldsymbol{z}_\ell - \eta \cdot \boldsymbol{\sigma}\Big([\boldsymbol{C}_\ell^1 \boldsymbol{z}_\ell, \ldots, \boldsymbol{C}_\ell^k \boldsymbol{z}_\ell]\Big) \quad \text{subject to} \quad \boldsymbol{z}_{\ell+1} \in \mathbb{S}^{n-1}, \qquad (10)$$

with the nonlinear function $\boldsymbol{\sigma}(\cdot)$ defined above and $\boldsymbol{\theta}_\ell$ collecting all the layer-wise parameters including $\boldsymbol{E}_\ell, \boldsymbol{C}_\ell^j, \gamma_j$ and $\lambda$, and with features at each layer always "normalized" onto a sphere $\mathbb{S}^{n-1}$, denoted as $\mathcal{P}_{\mathbb{S}^{n-1}}$. The form of increment in (10) can be illustrated by a diagram in Figure 1.

**Deep Network from Rate Reduction.** Notice that the increment is constructed to emulate the gradient ascent for the rate reduction $\Delta R$. Hence by transforming the features iteratively via the above process, we expect the rate reduction to increase, as we will see in the experimental section. This iterative process, once converged say after $L$ iterations, gives the desired feature map $f(\boldsymbol{x}, \boldsymbol{\theta})$ on the input $\boldsymbol{z}_0 = \boldsymbol{x}$, precisely in the form of a *deep network*, in which each layer has the structure shown in Figure 1:

$$f(\boldsymbol{x}, \boldsymbol{\theta}) = \phi^L \circ \phi^{L-1} \circ \cdots \circ \phi^0(\boldsymbol{x}), \quad \text{with} \quad \phi^\ell(\boldsymbol{z}_\ell, \boldsymbol{\theta}_\ell) \doteq \mathcal{P}_{\mathbb{S}^{n-1}}[\boldsymbol{z}_\ell + \eta \cdot g(\boldsymbol{z}_\ell, \boldsymbol{\theta}_\ell)]. \qquad (11)$$

As this deep network is derived from maximizing the rate **redu**ction, we call it the **ReduNet**. Notice that all parameters of the network are explicitly constructed layer by layer in a *forward propagation* fashion. Once constructed, there is no need of any additional supervised learning, say via back propagation. As suggested in Yu et al. (2020), the so learned features can be directly used for classification via a nearest subspace classifier.

**Comparison with Other Approaches and Architectures.** Structural similarities between deep networks and iterative optimization schemes, especially those for solving sparse coding, have been long noticed. In particular, Gregor & LeCun (2010) has argued that algorithms for sparse coding, such as the FISTA algorithm (Beck & Teboulle, 2009), can be viewed as a deep network and be trained for better coding performance, known as LISTA. Later Monga et al. (2019); Sun et al. (2020) have proposed similar interpretation of deep networks as unrolling algorithms for sparse coding. Like all networks that are inspired by unfolding certain iterative optimization schemes, the structure of the ReduNet naturally contains a skip connection between adjacent layers as in the ResNet (He

---

[4]The choice of the softmax is mostly for its simplicity as it is widely used in other (forward components of) deep networks for purposes such as selection, gating (Shazeer et al., 2017) and routing (Sabour et al., 2017). In principle, this term can be approximated by other operators, say using ReLU that is more amenable to training with back propagation, see Remark 3 in Appendix A.

et al., 2016). Remark 4 in Appendix A discusses possible improvement to the basic gradient scheme that may introduce additional skip connections beyond adjacent layers. The remaining $k+1$ parallel channels $\boldsymbol{E}, \boldsymbol{C}^j$ of the ReduNet actual draw resemblance to the parallel structures that people later found empirically beneficial for deep networks, e.g. ResNEXT (Xie et al., 2017) or the mixture of experts (MoE) module adopted in Shazeer et al. (2017). But a major difference here is that all components (layers, channels, and operators) of the ReduNet are by explicit construction from first principles and they all have precise optimization, statistical and geometric interpretation. Furthermore, there is no need to learn them from back-propagation, although in principle one still could if further fine-tuning of the network is needed (see Remark 3 of Appendix A for more discussions).

## 2.3 Deep Convolution Networks from Shift-Invariant Rate Reduction

We next examine ReduNet from the perspective of invariance to transformation. Using the basic and important case of shift/translation invariance as an example, we will show that for data which are compatible with an invariant classifier, the ReduNet construction automatically takes the form of *a (multi-channel) convolutional neural network*, rather than heuristically imposed upon.

**1D Serial Data and Shift Invariance.** For one-dimensional data $\boldsymbol{x} \in \mathbb{R}^n$ under shift symmetry, we take $\mathbb{G}$ to be the group of circular shifts. Each observation $\boldsymbol{x}^i$ generates a family $\{\boldsymbol{x}^i \circ \mathfrak{g} \,|\, \mathfrak{g} \in \mathbb{G}\}$ of shifted copies, which are the columns of the circulant matrix $\mathsf{circ}(\boldsymbol{x}^i) \in \mathbb{R}^{n \times n}$ (see Appendix B.1 or Kra & Simanca (2012) for properties of circulant matrices).

What happens if we construct the ReduNet from these families $\boldsymbol{Z}_1 = [\mathsf{circ}(\boldsymbol{x}^1), \ldots, \mathsf{circ}(\boldsymbol{x}^m)]$? The data covariance matrix:

$$\boldsymbol{Z}_1 \boldsymbol{Z}_1^* = \begin{bmatrix} \mathsf{circ}(\boldsymbol{x}^1), \ldots, \mathsf{circ}(\boldsymbol{x}^m) \end{bmatrix} \begin{bmatrix} \mathsf{circ}(\boldsymbol{x}^1), \ldots, \mathsf{circ}(\boldsymbol{x}^m) \end{bmatrix}^* = \sum_{i=1}^m \mathsf{circ}(\boldsymbol{x}^i) \mathsf{circ}(\boldsymbol{x}^i)^* \in \mathbb{R}^{n \times n}$$

associated with this family of samples is *automatically* a (symmetric) circulant matrix. Moreover, because the circulant property is preserved under sums, inverses, and products, the matrices $\boldsymbol{E}_1$ and $\boldsymbol{C}_1^j$ are also automatically circulant matrices, whose application to a feature vector $\boldsymbol{z}$ can be implemented using cyclic convolution "$\circledast$" (see Proposition B.1 of Appendix B):

$$\boldsymbol{z}_2 \propto \boldsymbol{z}_1 + \eta \cdot g(\boldsymbol{z}_1, \boldsymbol{\theta}_1) = \boldsymbol{z}_1 + \eta \cdot \boldsymbol{e}_1 \circledast \boldsymbol{z}_1 - \eta \cdot \boldsymbol{\sigma} \Big( [\boldsymbol{c}_1^1 \circledast \boldsymbol{z}_1, \ldots, \boldsymbol{c}_1^k \circledast \boldsymbol{z}_1] \Big). \tag{12}$$

Because $g(\cdot, \boldsymbol{\theta}_1)$ consists only of operations that co-vary with cyclic shifts, the features $\boldsymbol{Z}_2$ at the next level again consist of families of shifts: $\boldsymbol{Z}_2 = \big[\mathsf{circ}(\boldsymbol{x}^1 + \eta g(\boldsymbol{x}^1, \boldsymbol{\theta}_1)), \ldots, \mathsf{circ}(\boldsymbol{x}^m + \eta g(\boldsymbol{x}^m, \boldsymbol{\theta}_m))\big]$. Continuing inductively, we see that all matrices $\boldsymbol{E}_\ell$ and $\boldsymbol{C}_\ell^j$ based on such $\boldsymbol{Z}_\ell$ are circulant. By virtue of the properties of the data, ReduNet has taken the form of a convolutional network, *with no need to explicitly choose this structure!*

**The Role of Multiple Channels and Sparsity.** There is one problem though: In general, the set of all circular permutations of a vector $\boldsymbol{z}$ give a full-rank matrix. That is, the $n$ "augmented" features associated with each sample (hence each class) typically already span the entire space $\mathbb{R}^n$. The $\mathrm{MCR}^2$ objective (1) will not be able to distinguish classes as different subspaces.

One natural remedy is to improve the separability of the data by "lifting" the features to a higher dimensional space, e.g., by taking their responses to multiple, filters $\boldsymbol{k}_1, \ldots, \boldsymbol{k}_C \in \mathbb{R}^n$:

$$\boldsymbol{z}[c] = \boldsymbol{k}_c \circledast \boldsymbol{z} = \mathsf{circ}(\boldsymbol{k}_c) \boldsymbol{z} \quad \in \mathbb{R}^n, \quad c = 1, \ldots, C. \tag{13}$$

The filers can be pre-designed invariance-promoting filters,[5] or adaptively learned from the data,[6] or randomly selected as we do in our experiments. This operation lifts each original feature vector $\boldsymbol{z} \in \mathbb{R}^n$ to a $C$-channel feature, denoted $\bar{\boldsymbol{z}} \doteq [\boldsymbol{z}[1], \ldots, \boldsymbol{z}[C]]^* \in \mathbb{R}^{C \times n}$. If we stack the multiple channels of a feature $\bar{\boldsymbol{z}}$ as a column vector $\mathsf{vec}(\bar{\boldsymbol{z}}) \in \mathbb{R}^{nC}$, the associated circulant version $\mathsf{circ}(\bar{\boldsymbol{z}})$ and its data covariance matrix, denoted as $\bar{\boldsymbol{\Sigma}}$, for all its shifted versions are given as:

$$\mathsf{circ}(\bar{\boldsymbol{z}}) \doteq \begin{bmatrix} \mathsf{circ}(\boldsymbol{z}[1]) \\ \vdots \\ \mathsf{circ}(\boldsymbol{z}[C]) \end{bmatrix} \in \mathbb{R}^{nC \times n}, \quad \bar{\boldsymbol{\Sigma}} \doteq \begin{bmatrix} \mathsf{circ}(\boldsymbol{z}[1]) \\ \vdots \\ \mathsf{circ}(\boldsymbol{z}[C]) \end{bmatrix} \big[ \mathsf{circ}(\boldsymbol{z}[1])^*, \ldots, \mathsf{circ}(\boldsymbol{z}[C])^* \big] \in \mathbb{R}^{nC \times nC}, \tag{14}$$

---

[5] For 1D signals like audio, one may consider the conventional short time Fourier transform (STFT); for 2D images, one may consider 2D wavelets as in the ScatteringNet (Bruna & Mallat, 2013).

[6] For learned filters, one can learn filters as the principal components of samples as in the PCANet (Chan et al., 2015) or from convolution dictionary learning (Li & Bresler, 2019; Qu et al., 2019).

where $\text{circ}(\boldsymbol{z}[c]) \in \mathbb{R}^{n \times n}$ with $c \in [C]$ is the circulant version of the $c$-th channel of the feature $\bar{\boldsymbol{z}}$. Then the columns of $\text{circ}(\bar{\boldsymbol{z}})$ will only span at most an $n$-dimensional proper subspace in $\mathbb{R}^{nC}$.

However, this operation does not yet render the classes separable – features associated with other classes will span the *same* $n$-dimensional subspace. This reflects a fundamental conflict between linear (subspace) modeling and invariance. One way of resolving this conflict is to leverage additional structure within each class, in the form of *sparsity*: signals within each class can be assumed to be generated not as arbitrary linear combinations of basis vectors, but as sparse combinations of atoms of different (incoherent) dictionaries $\mathcal{D}_j$. Under this assumption, if the convolution kernels $\{\boldsymbol{k}_c\}$ match well the sparsifying dictionaries,[7] the multi-channel responses should be sparse. Hence we may take an entry-wise *sparsity-promoting nonlinear thresholding*, say $\boldsymbol{\tau}(\cdot)$, on the filter outputs by setting small (say absolute value below $\epsilon$) or negative responses to be zero:[8]

$$\bar{\boldsymbol{z}} = \boldsymbol{\tau}\big[\text{circ}(\boldsymbol{k}_1)\boldsymbol{z}, \ldots, \text{circ}(\boldsymbol{k}_C)\boldsymbol{z}\big] \quad \in \mathbb{R}^{n \times C}. \tag{15}$$

These features can be assumed to lie on a lower-dimensional (nonlinear) submanifold of $\mathbb{R}^{n \times C}$, which can be linearized and separated from the other classes by subsequent ReduNet layers.

This multi-channel ReduNet retains the good invariance properties described above: all $\boldsymbol{E}_\ell$ and $\boldsymbol{C}_\ell^j$ matrices are block circulant, and represent *multi-channel 1D circular convolutions* (see Proposition B.2 of Appendix B for a rigorous statement and proof):

$$\bar{\boldsymbol{E}}(\bar{\boldsymbol{z}}) = \bar{\boldsymbol{e}} \circledast \bar{\boldsymbol{z}}, \quad \bar{\boldsymbol{C}}^j(\bar{\boldsymbol{z}}) = \bar{\boldsymbol{c}}^j \circledast \bar{\boldsymbol{z}} \quad \in \mathbb{R}^{n \times C}, \quad j = 1, \ldots, k, \tag{16}$$

where $\bar{\boldsymbol{e}}, \bar{\boldsymbol{c}}^j \in \mathbb{R}^{C \times C \times n}$. Hence by construction, the resulting ReduNet is a deep convolutional network for multi-channel 1D signals. Unlike Xception nets Chollet (2017), these multi-channel convolutions in general are *not* depthwise separable.[9]

**Fast Computation in the Spectral Domain.** Since all circulant matrices can be simultaneously diagonalized by the discrete Fourier transform matrix[10] $\boldsymbol{F}$: $\text{circ}(\boldsymbol{z}) = \boldsymbol{F}^* \boldsymbol{D} \boldsymbol{F}$ (see Fact 5 in Appendix B.1), all $\bar{\boldsymbol{\Sigma}}$ of the form (14) can be converted to a standard "blocks of diagonals" form:

$$\bar{\boldsymbol{\Sigma}} = \begin{bmatrix} \boldsymbol{F}^* & \boldsymbol{0} & \boldsymbol{0} \\ \boldsymbol{0} & \ddots & \boldsymbol{0} \\ \boldsymbol{0} & \boldsymbol{0} & \boldsymbol{F}^* \end{bmatrix} \begin{bmatrix} \boldsymbol{D}_{11} & \cdots & \boldsymbol{D}_{1C} \\ \vdots & \ddots & \vdots \\ \boldsymbol{D}_{C1} & \cdots & \boldsymbol{D}_{CC} \end{bmatrix} \begin{bmatrix} \boldsymbol{F} & \boldsymbol{0} & \boldsymbol{0} \\ \boldsymbol{0} & \ddots & \boldsymbol{0} \\ \boldsymbol{0} & \boldsymbol{0} & \boldsymbol{F} \end{bmatrix} \quad \in \mathbb{R}^{nC \times nC}, \tag{17}$$

where each block $\boldsymbol{D}_{kl}$ is an $n \times n$ diagonal matrix. The middle of RHS of (17) is a block diagonal matrix after a permutation of rows and columns. Hence, to compute $\bar{\boldsymbol{E}}$ and $\bar{\boldsymbol{C}}^j \in \mathbb{R}^{nC \times nC}$, we only have to compute in the frequency domain the inverse of $C \times C$ blocks for $n$ times and the overall complexity would be $O(nC^3)$ instead of $O((nC)^3)$ for inverting a generic $nC \times nC$ matrix.[11] More details for implementing the network in the spectral domain can be found in Appendix B.3 (see Theorem B.3 for a rigorous statement and Algorithm 1 for implementation details).

**Connections to Recurrent and Convolutional Sparse Coding.** The sparse coding perspective of Gregor & LeCun (2010) is later extended to recurrent and convolutional networks for serial data, e.g. Wisdom et al. (2016); Papyan et al. (2016); Sulam et al. (2018); Monga et al. (2019). Although both sparsity and convolution are advocated as desired characteristics for deep networks, they do not explicitly justify the necessity of sparsity and convolutions from the objective of the network, say classification. In our framework, we see how multi-channel convolutions ($\bar{\boldsymbol{E}}, \bar{\boldsymbol{C}}^j$), different nonlinear activations ($\hat{\boldsymbol{\pi}}^j, \boldsymbol{\tau}$), and the sparsity requirement are derived *from*, rather than heuristically proposed for, the objective of maximizing rate reduction of the features while enforcing shift invariance.

**2D Images and Translation Invariance.** In the case of classifying images invariant to arbitrary 2D translation, we may view the image (feature) $\boldsymbol{z} \in \mathbb{R}^{(W \times H) \times C}$ as a function defined on a torus $\mathcal{T}^2$ (discretized as a $W \times H$ grid) and consider $\mathbb{G}$ to be the (Abelian) group of all 2D (circular)

---

[7]There is a vast literature on how to learn the most compact and optimal sparsifying dictionaries from sample data, e.g. (Li & Bresler, 2019; Qu et al., 2019). Nevertheless, in practice, often a sufficient number of random filters suffice the purpose of ensuring features of different classes are separable (Chan et al., 2015).

[8]Here the nonlinear operator $\boldsymbol{\tau}$ can be chosen to be a soft thresholding or a ReLU.

[9]It remains open what additional structures on the data would lead to depthwise separable convolutions.

[10]Here we scaled the matrix $\boldsymbol{F}$ to be unitary, hence it differs from the conventional DFT matrix by a $1/\sqrt{n}$.

[11]There are strong scientific evidences that neurons in the visual cortex encode and transmit information in the rate of spiking, hence the so-called spiking neurons (Softky & Koch, 1993; Eliasmith & Anderson, 2003). Nature might be exploiting the computational efficiency in the frequency domain for achieving shift invariance.

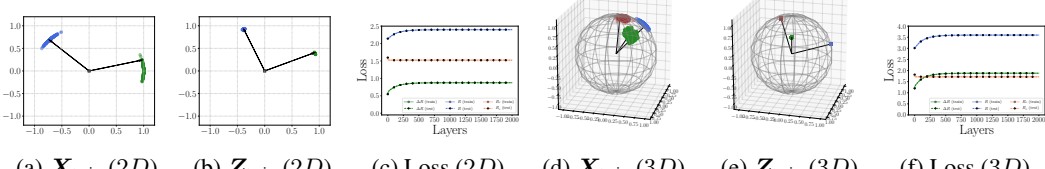

(a) $\boldsymbol{X}_{\text{train}}$ (2D)    (b) $\boldsymbol{Z}_{\text{train}}$ (2D)    (c) Loss (2D)    (d) $\boldsymbol{X}_{\text{train}}$ (3D)    (e) $\boldsymbol{Z}_{\text{train}}$ (3D)    (f) Loss (3D)

Figure 2: Original samples and learned representations for 2D and 3D Mixture of Gaussians. We visualize data points $\boldsymbol{X}$ (before mapping) and features $\boldsymbol{Z}$ (after mapping) by scatter plot. In each scatter plot, each color represents one class of samples. We also show the plots for the progression of values of the objective functions.

translations on the torus. As we will show in the Appendix C, the associated linear operators $\bar{\boldsymbol{E}}$ and $\bar{\boldsymbol{C}}^j$'s act on the image feature $\boldsymbol{z}$ as *multi-channel 2D circular convolutions*. The resulting network will be a deep convolutional network that shares the same multi-channel convolution structures as conventional CNNs for 2D images (LeCun et al., 1995; Krizhevsky et al., 2012). The difference is that, again, the architectures and parameters of our network are derived from the rate reduction objective, and so are the nonlinear activation $\widehat{\boldsymbol{\pi}}^j$ and $\boldsymbol{\tau}$. Again, our derivation in Appendix C shows that this multi-channel 2D convolutional network can be constructed more efficiently in the spectral domain (see Theorem C.1 of Appendix C for a rigorous statement and justification).

## 3 EXPERIMENTS

We now *verify* whether the so constructed ReduNet achieves its design objectives through experiments on synthetic data and real images. The datasets and experiments are chosen to clearly demonstrate the behaviors of the network obtained by our algorithm, in terms of learning the correct discriminative representation and truly achieving invariance. It is not the purpose of this work to push the state of the art on any real datasets with highly engineered networks and systems, although we believe this framework has this potential in the future. All code is implemented in Python mainly using NumPy. All our experiments are conducted in a computer with 2.8 GHz Intel i7 CPU and 16GB of memory. Implementation details and more experiments and can be found in Appendix D.

**Learning Mixture of Gaussians in $\mathbb{S}^1$ and $\mathbb{S}^2$.** Consider a mixture of two Gaussian distributions in $\mathbb{R}^2$ that is projected onto $\mathbb{S}^1$. We first generate data points from these two distributions, $\boldsymbol{X}_1 = [\boldsymbol{x}_1^1, \ldots, \boldsymbol{x}_1^m] \in \mathbb{R}^{2 \times m}$, $\boldsymbol{x}_1^i \sim \mathcal{N}(\boldsymbol{\mu}_1, \sigma_1 \boldsymbol{I})$, and $\boldsymbol{\pi}(\boldsymbol{x}_1^i) = 1$; $\boldsymbol{X}_2 = [\boldsymbol{x}_2^1, \ldots, \boldsymbol{x}_2^m] \in \mathbb{R}^{2 \times m}$, $\boldsymbol{x}_2^i \sim \mathcal{N}(\boldsymbol{\mu}_2, \sigma_2 \boldsymbol{I})$, and $\boldsymbol{\pi}(\boldsymbol{x}_2^i) = 2$. We set $m = 500, \sigma_1 = \sigma_2 = 0.1$ and $\boldsymbol{\mu}_1, \boldsymbol{\mu}_2 \in \mathbb{S}^1$. Then we project all the data points onto $\mathbb{S}^1$, i.e., $\boldsymbol{x}_j^i / \|\boldsymbol{x}_j^i\|_2$. To construct the network (computing $\boldsymbol{E}, \boldsymbol{C}^j$ for each layer), we set the # of iterations/layers $L = 2,000$,[12] step size $\eta = 0.5$, and precision $\epsilon = 0.1$. As shown in Figure 2a-2b, we can observe that after the mapping $f(\cdot, \boldsymbol{\theta})$, samples from the same class converge to a single cluster and the angle between two different clusters is approximately $\pi/2$, which is well aligned with the optimal solution $\boldsymbol{Z}_\star$ of the MCR$^2$ loss in $\mathbb{S}^1$. MCR$^2$ loss of features on different layers can be found in Figure 2c. Empirically, we find that our constructed network is able to maximize MCR$^2$ loss and converges stably. Similarly, we consider mixture of three Gaussian distributions in $\mathbb{R}^3$ with means $\boldsymbol{\mu}_1, \boldsymbol{\mu}_2, \boldsymbol{\mu}_3$ uniformly in $\mathbb{S}^2$, and variance $\sigma_1 = \sigma_2 = \sigma_3 = 0.1$, and all data points are projected onto $\mathbb{S}^2$ (See Figure 2d-2f). We can observe similar behavior as in $\mathbb{S}^2$, i.e., samples from the same class converge to one cluster and different clusters are orthogonal to each other. Moreover, we sample new data points from the same distributions for both cases and find that new samples form the same class consistently converge to the same cluster as the training samples. More examples and details can be found in Appendix D.

**Learning Shift Invariant Features.** As described in § 2.3, by maximizing the rate reduction via Eq. (12), we are able to explicitly construct operators that are invariant to (circular) shifts. To verify the effectiveness of our proposed network on shift invariance tasks, we apply our network to classify signals sampled from two different 1D functions. The underlying function of the first class is sinusoidal signal $h_1(t) = \sin(t) + \epsilon$, and the second class is a composition of sign and sin function, $h_2(t) = \text{sign}(\sin(t)) + \epsilon$, where $\epsilon \sim \mathcal{N}(0, 0.1)$. (See Figure 7 in Appendix D). Each sample is generated by first picking $t_0 \in [0, 10\pi]$, then obtaining $n$ equidistant point within the boundaries $[t_0, t_0 + 2\pi]$ with i.i.d Gaussian noise. Detailed implementations for sampling from $h_1$ and $h_2$ can be found in Appendix D.3. We generate a dataset which contains $m$ samples, with $m/2$ samples in each class, i.e., $\boldsymbol{X} = [\boldsymbol{X}_1, \boldsymbol{X}_2] \in \mathbb{R}^{n \times m}$. Then each sample is lifted to a $C$-

---

[12]It is remarkable to see how easily our framework leads to working deep networks with thousands of layers! But this also indicates the efficiency of the layers is not so high. Remark 4 provides possible ways to improve.

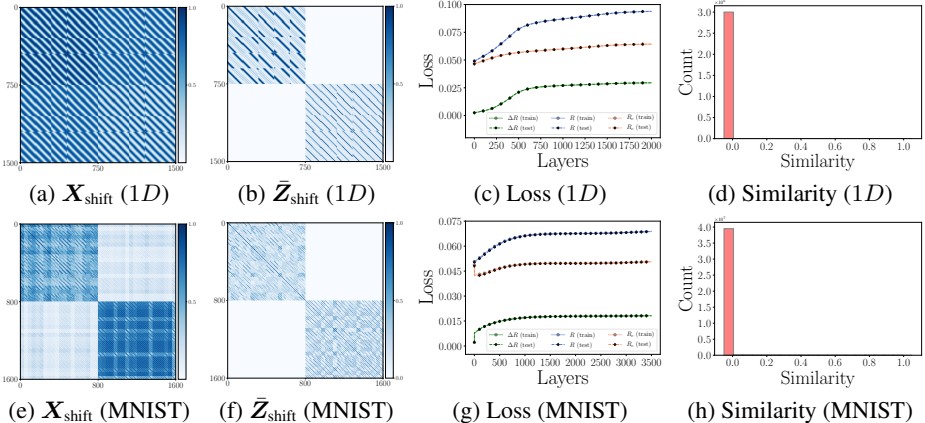

(a) $\boldsymbol{X}_{\mathrm{shift}}$ (1D)    (b) $\bar{\boldsymbol{Z}}_{\mathrm{shift}}$ (1D)    (c) Loss (1D)    (d) Similarity (1D)

(e) $\boldsymbol{X}_{\mathrm{shift}}$ (MNIST)    (f) $\bar{\boldsymbol{Z}}_{\mathrm{shift}}$ (MNIST)    (g) Loss (MNIST)    (h) Similarity (MNIST)

Figure 3: Heatmaps of cosine similarity between data $\boldsymbol{X}_{\mathrm{shift}}$/learned features $\bar{\boldsymbol{Z}}_{\mathrm{shift}}$, MCR$^2$ loss, and distance between shift samples and subspaces. For (a), (b), (e), (f), we pick one sample from each class and augment the sample with its every possible shifted ones, then calculate the cosine similarity between these augmented samples. For (d), (h), we first augment each samples in the dataset with its every possible shifted ones, then we evaluate the cosine similarity (in absolute value) between pairs across classes: for each pair, one sample is from training and one sample is from test which belong to different classes.

channel feature as defined in (13), i.e., $\bar{\boldsymbol{X}} \in \mathbb{R}^{(n \cdot C) \times m}$. For training data, We set the number of features $n = 150$, samples $m = 400$, channels $C = 7$, iterations/layers $L = 2,000$, step size $\eta = 0.1$, and precision $\epsilon = 0.1$. We sample the same number of test data points followed by the same procedure. As shown in Figure 8, we observe that the network can map the two classes of signals to orthogonal subspaces both on training and test datasets. To verify invariance property of the network, we first pick 5 signal samples from each class (from test dataset) and get their corresponding augmented samples by shifting. Then we have $m = 1,500$ augmented samples for each original signal, $\boldsymbol{X}_{\mathrm{shift}} \in \mathbb{R}^{150 \times 1,500}$, and we visualize the pairwise inner product of $\boldsymbol{X}_{\mathrm{shift}}$ and their representations, $\bar{\boldsymbol{Z}}_{\mathrm{shift}} \in \mathbb{R}^{(150 \cdot 7) \times 1,500}$ in Figure 3a-3b. Moreover, we augment every sample (from the test dataset) with its all possible shifted versions and calculate the cosine similarity between their representations and the representations of all the training samples from the other class (in Figure 3d). We find that the proposed network can map different classes of signals (including all shifted augmentations) to orthogonal subspaces, to increase the MCR$^2$ loss (shown in Figure 3c).

**Rotational Invariance on MNIST Digits.** We study the ReduNet on learning *rotation* invariant features on MNIST dataset (LeCun, 1998). We impose a polar grid on the image $\boldsymbol{x} \in \mathbb{R}^{H \times W}$, with its geometric center being the center of the 2D polar grid. For each radius $r_i$, $i \in [C]$, we can sample $\Gamma$ pixels with respect to each angle $\gamma_l = l \cdot (2\pi/\Gamma)$ with $l \in [\Gamma]$. Then given an image sample $\boldsymbol{x}$ from the dataset, we represent the image in a polar coordinate representation $\boldsymbol{x}(p) = (\gamma_{l,i}, r_{l,i}) \in \mathbb{R}^{\Gamma \times C}$. Our goal is to learn rotation invariant features, i.e., we expect to learn $f(\cdot, \boldsymbol{\theta})$ such that $\{f(\boldsymbol{x}(p) \circ \mathfrak{g}, \boldsymbol{\theta})\}_{\mathfrak{g} \in \mathbb{G}}$ lie in the same subspace, where $\mathfrak{g}$ is the shift transformation in polar angle. By performing polar coordinate transformation for images from digit '0' and digit '1' in the training dataset, we can obtain the data matrix $\boldsymbol{X}(p) \in \mathbb{R}^{(\Gamma \cdot C) \times m}$. We use $m = 2,000$ training samples, set $\Gamma = 200$ and $C = 5$ for polar transformation, and set iteration $L = 3,500$, precision $\epsilon = 0.1$, step-size $\eta = 0.5$. We generate $1,000$ test samples followed by the same procedure. In Figure 10, we can see that our proposed ReduNet is able to map most samples from different classes to orthogonal subspaces (w.r.t. class) on test dataset. Meanwhile, in Figure 3e, 3f, and 3h, we observe that the learnt features are invariant to shift transformation in polar angle (i.e., arbitrary rotation in $\boldsymbol{x}$).

## 4 CONCLUSIONS AND FUTURE WORK

This work offers an interpretation of deep (convolutional) networks by construction from first principles. It provides a rigorous explanation for the deep architecture and components from the perspective of optimizing the rate reduction objective. Simulations and experiments on basic data sets clearly verify the so-constructed ReduNet achieves the desired functionality and objective. Although in this work the ReduNet is forward constructed, one may study how to effectively fine tune it via back propagation. We believe rate reduction provides a principled framework for designing new networks with interpretable architectures and operators that can scale up to real-world datasets and problems, with better performance guarantees. This framework can also be naturally extended to settings of *online* or *unsupervised* learning if $\boldsymbol{\Pi}$ is partially or not known and is to be optimized.

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

## A   ADDITIONAL REMARKS AND EXTENSIONS

**Remark 1 (Interpretation of the Two Linear Operators)**   *For any $\boldsymbol{z}_\ell$ we have*

$$(\boldsymbol{I} + \alpha \boldsymbol{Z}_\ell \boldsymbol{Z}_\ell^*)^{-1} \boldsymbol{z}_\ell = \boldsymbol{z}_\ell - \boldsymbol{Z}_\ell \boldsymbol{q}_\ell^* \quad where \quad \boldsymbol{q}_\ell^* \doteq \operatorname*{argmin}_{\boldsymbol{q}_\ell} \alpha \|\boldsymbol{z}_\ell - \boldsymbol{Z}_\ell \boldsymbol{q}_\ell\|_2^2 + \|\boldsymbol{q}_\ell\|_2^2. \qquad (18)$$

*Notice that $\boldsymbol{q}_\ell^*$ is exactly the solution to the ridge regression by all the data points $\boldsymbol{Z}_\ell$ concerned. Therefore, $\boldsymbol{E}_\ell$ (similarly for $\boldsymbol{C}_\ell^j$) is approximately (i.e. when m is large enough) the projection onto the orthogonal complement of the subspace spanned by columns of $\boldsymbol{Z}_\ell$. Another way to interpret the matrix $\boldsymbol{E}_\ell$ is through eigenvalue decomposition of the covariance matrix $\boldsymbol{Z}_\ell \boldsymbol{Z}_\ell^*$. Assuming that $\boldsymbol{Z}_\ell \boldsymbol{Z}_\ell^* \doteq \boldsymbol{U}_\ell \boldsymbol{\Lambda}_\ell \boldsymbol{U}_\ell^*$ where $\boldsymbol{\Lambda}_\ell \doteq \mathsf{diag}\{\sigma_1, \ldots, \sigma_d\}$, we have*

$$\boldsymbol{E}_\ell = \alpha \, \boldsymbol{U}_\ell \, \mathsf{diag} \left\{ \frac{1}{1 + \alpha \sigma_1}, \ldots, \frac{1}{1 + \alpha \sigma_d} \right\} \boldsymbol{U}_\ell^*. \qquad (19)$$

*Therefore, the matrix $\boldsymbol{E}_\ell$ operates on a vector $\boldsymbol{z}_\ell$ by stretching in a way that directions of large variance are shrunk while directions of vanishing variance are kept. These are exactly the directions (4) in which we move the features so that the overall volume expands and the coding rate will increase, hence the positive sign. To the opposite effect, the directions associated with (5) are exactly "residuals" of features of each class deviate from the subspace to which they are supposed to belong. These are exactly the directions in which the features need to be compressed back onto their respective subspace, hence the negative sign.*

*Essentially, all linear operations in the ReduNet are determined by data conducting "auto-regressions" among themselves. The recent renewed understanding about ridge regression in an over-parameterized setting (Yang et al., 2020) indicates that using seemingly redundantly sampled data (from each subspaces) as regressors do not lead to overfitting.*

**Remark 2 (Connection and Difference from Neural ODE)**   *Notice that one may interpret the increment (7) as a discretized version of a continuous ordinary differential equation (ODE):*

$$\dot{\boldsymbol{z}} = g(\boldsymbol{z}, \theta). \qquad (20)$$

*Hence the (deep) network so constructed can be interpreted as certain neural ODE (Chen et al., 2018). Nevertheless, unlike neural ODE where the flow $g$ is chosen to be some generic structures, here our $g(\boldsymbol{z}, \theta)$ is to emulate the gradient flow of the rate reduction on the feature set*

$$\dot{\boldsymbol{Z}} = \frac{\partial \Delta R}{\partial \boldsymbol{Z}}, \qquad (21)$$

*and its structure is entirely derived and fully determined from this objective, without any other priors or heuristics.*

**Remark 3 (Approximate with a ReLU Network)**   *In practice, there are many other simpler nonlinear activation functions that one can use to approximate the membership $\widehat{\boldsymbol{\pi}}(\cdot)$ and subsequently the nonlinear operation $\boldsymbol{\sigma}$ in (9). Notice that the geometric meaning of $\boldsymbol{\sigma}$ in (9) is to compute the "residual" of each feature against the subspace to which it belongs. So when we restrict all our features to be in the first (positive) quadrant of the feature space,[13] one may approximate this residual using the rectified linear units operation, ReLUs, on $\boldsymbol{p}_j = \boldsymbol{C}_\ell^j \boldsymbol{z}_\ell$ or its orthogonal complement:*

$$\boldsymbol{\sigma}(\boldsymbol{z}_\ell) \, \propto \, \boldsymbol{z}_\ell - \sum_{j=1}^{k} ReLU\big(\boldsymbol{P}_\ell^j \boldsymbol{z}_\ell\big), \qquad (22)$$

*where $\boldsymbol{P}_\ell^j = (\boldsymbol{C}_\ell^j)^\perp$ is the projection onto the $j$-th class[14] and $ReLU(x) = \max(0, x)$. The above approximation is good under the more restrictive assumption that projection of $\boldsymbol{z}_\ell$ on the correct class via $\boldsymbol{P}_\ell^j$ is mostly large and positive and yet small or negative for other classes.*

*The resulting ReduNet will be a network primarily involving ReLU operations and feature normalization (onto $\mathbb{S}^{n-1}$) between each layer. Although in this work, we have argued that the forward-constructed ReduNet network already works to a large extent, in practice one certainly can conduct back-propagation to further fine tune the so-obtained network, say to correct some remaining errors in predicting labels of the training data. Empirically, people have found that deep networks with ReLU activations are easier to train via back propagation (Krizhevsky et al., 2012).*

---

[13]Most current neural networks seem to adopt this regime.

[14]$\boldsymbol{P}_\ell^j$ can be viewed as the orthogonal complement to $\boldsymbol{C}_\ell^j$.

**Remark 4 (Accelerated Optimization via Additional Skip Connections)** *Empirically, people have found that additional skip connections across multiple layers may improve the network performance, e.g. the DenseNet (Huang et al., 2017). In our framework, the role of each layer is precisely interpreted as one iterative gradient ascent step for the objective function $\Delta R$. In our experiments (see Section 3), we have observed that the basic gradient scheme sometimes converges slowly, resulting in deep networks with thousands of layers (iterations)! To improve the efficiency of the basic ReduNet, one may consider in the future accelerated gradient methods such as the Nesterov acceleration (Nesterov, 1983) or perturbed accelerated gradient descent (Jin et al., 2018). Say to minimize or maximize a function $h(\boldsymbol{z})$, such accelerated methods usually take the form:*

$$\begin{cases} \boldsymbol{p}_{\ell+1} & = \quad \boldsymbol{z}_\ell + \beta_\ell \cdot (\boldsymbol{z}_\ell - \boldsymbol{z}_{\ell-1}), \\ \boldsymbol{z}_{\ell+1} & = \quad \boldsymbol{p}_{\ell+1} + \eta \cdot \nabla h(\boldsymbol{p}_{\ell+1}). \end{cases} \tag{23}$$

*Hence they require introducing additional skip connections among three layers $\ell-1$, $\ell$ and $\ell+1$. For typical convex or nonconvex programs, the above accelerated schemes can often reduce the number of iterations by a magnitude.*

# B  1D CIRCULAR SHIFT INVARIANCE

It has been long known that to implement a convolutional neural network, one can achieve higher computational efficiency by implementing the network in the spectral domain via the fast Fourier transform (Mathieu et al., 2013; Lavin & Gray, 2016; Vasilache et al., 2015). However, our purpose here is different: We want to show that the linear operators $E$ and $C^j$ derived from the gradient flow of MCR$^2$ are naturally convolutions when we enforce shift-invariance rigorously. Their convolution structure is derived from the rate reduction objective, rather than heuristically imposed upon the network. Furthermore, the computation involved in constructing these linear operators has a naturally efficient implementation in the spectral domain via fast Fourier transform. Arguably this work is the first to show multi-channel convolutions, together with other convolution-preserving nonlinear operations in the ReduNet, are both necessary and sufficient to ensure shift invariance.

To be somewhat self-contained and self-consistent, in this section, we first introduce our notation and review some of the key properties of circulant matrices which will be used to characterize the properties of the linear operators $E$ and $C^j$ and to compute them efficiently. The reader may refer to Kra & Simanca (2012) for a more rigorous exposition on circulant matrices.

## B.1  PROPERTIES OF CIRCULANT MATRIX AND CIRCULAR CONVOLUTION

Given a vector $z = [z_0, z_1, \ldots, z_{n-1}]^* \in \mathbb{R}^n$,[15] we may arrange all its circular shifted versions in a circulant matrix form as

$$\mathsf{circ}(z) \doteq \begin{bmatrix} z_0 & z_{n-1} & \cdots & z_2 & z_1 \\ z_1 & z_0 & z_{n-1} & \cdots & z_2 \\ \vdots & z_1 & z_0 & \ddots & \vdots \\ z_{n-2} & \vdots & \ddots & \ddots & z_{n-1} \\ z_{n-1} & z_{n-2} & \cdots & z_1 & z_0 \end{bmatrix} \in \mathbb{R}^{n \times n}. \tag{24}$$

**Fact 1 (Convolution as matrix multiplication via circulant matrix)** *The multiplication of a circulant matrix* $\mathsf{circ}(z)$ *with a vector* $x \in \mathbb{R}^n$ *gives a circular (or cyclic) convolution, i.e.,*

$$\mathsf{circ}(z) \cdot x = z \circledast x, \tag{25}$$

*where*

$$(z \circledast x)_i = \sum_{j=0}^{n-1} x_j z_{i+n-j \bmod n}. \tag{26}$$

**Fact 2 (Properties of circulant matrices)** *Circulant matrices have the following properties:*

- *Transpose of a circulant matrix, say* $\mathsf{circ}(z)^*$, *is circulant;*

- *Multiplication of two circulant matrices is circulant, for example* $\mathsf{circ}(z)\mathsf{circ}(z)^*$;

- *For a non-singular circulant matrix, its inverse is also circulant (hence representing a circular convolution).*

These properties of circulant matrices are extensively used in this work as for characterizing the convolution structures of the operators $E$ and $C^j$.

Given a set of vectors $[z^1, \ldots, z^m] \in \mathbb{R}^{n \times m}$, let $\mathsf{circ}(z^i) \in \mathbb{R}^{n \times n}$ be the circulant matrix for $z^i$. Then we have the following:

**Proposition B.1 (Convolution structures of $E$ and $C^j$)** *Given a set of vectors* $Z = [z^1, \ldots, z^m]$, *the matrix:*

$$E = \alpha\big(I + \alpha \sum_{i=1}^{m} \mathsf{circ}(z^i)\mathsf{circ}(z^i)^*\big)^{-1}$$

*is a circulant matrix and represents a circular convolution:*

$$Ez = e \circledast z,$$

*where* $e$ *is the first column vector of* $E$. *Similarly, the matrices* $C^j$ *associated with any subsets of* $Z$ *are also circular convolutions.*

---

[15] We use superscript $^*$ to indicate (conjugate) transpose of a vector or a matrix

### B.2 CIRCULANT MATRIX AND CIRCULANT CONVOLUTION FOR MULTI-CHANNEL SIGNALS

In the remainder of this section, we view $z$ as a 1D signal such as an audio signal. Since we will deal with the more general case of multi-channel signals, we will use the traditional notation $T$ to denote the temporal length of the signal and $C$ for the number of channels. Conceptually, the "dimension" $n$ of such a multi-channel signal, if viewed as a vector, should be $n = CT$.[16] As we will also reveal additional interesting structures of the operators $E$ and $C^j$ in the spectral domain, we use $t$ as the index for time, $p$ for the index of frequency, and $c$ for the index of channel.

Given a multi-channel 1D signal $\bar{z} \in \mathbb{R}^{C \times T}$, we denote

$$\bar{z} = \begin{bmatrix} \bar{z}[1]^* \\ \vdots \\ \bar{z}[C]^* \end{bmatrix} = [\bar{z}(0), \bar{z}(1), \dots, \bar{z}(T-1)] = \{\bar{z}[c](t)\}_{c=1,t=0}^{c=C,t=T-1}. \tag{27}$$

To compute the coding rate reduction for a collection of such multi-channel 1D signals, we may flatten the matrix representation into a vector representation by stacking the multiple channels of $\bar{z}$ as a column vector. In particular, we let

$$\mathsf{vec}(\bar{z}) = [\bar{z}[1](0), \bar{z}[1](1), \dots, \bar{z}[1](T-1), \bar{z}[2](0), \dots] \quad \in \mathbb{R}^{(C \times T)}. \tag{28}$$

Furthermore, to obtain shift invariance for the coding rate reduction, we may generate a collection of shifted copies of $\bar{z}$ (along the temporal dimension). Stacking the vector representations for such shifted copies as column vectors, we obtain

$$\mathsf{circ}(\bar{z}) \doteq \begin{bmatrix} \mathsf{circ}(\bar{z}[1]) \\ \vdots \\ \mathsf{circ}(\bar{z}[C]) \end{bmatrix} \quad \in \mathbb{R}^{(C \times T) \times T}. \tag{29}$$

In above, we overload the notation "$\mathsf{circ}(\cdot)$" defined in (24).

We now consider a collection of $m$ multi-channel 1D signals $\{\bar{z}^i \in \mathbb{R}^{C \times T}\}_{i=1}^m$. Compactly representing the data by $\bar{Z} \in \mathbb{R}^{C \times T \times m}$ in which the $i$-th slice on the last dimension is $\bar{z}^i$, we denote

$$\bar{Z}[c] = [\bar{z}^1[c], \dots, \bar{z}^m[c]] \in \mathbb{R}^{T \times m}, \qquad \bar{Z}(t) = [\bar{z}^1(t), \dots, \bar{z}^m(t)] \in \mathbb{R}^{C \times m}. \tag{30}$$

In addition, we denote

$$\begin{aligned} \mathsf{vec}(\bar{Z}) &= [\mathsf{vec}(\bar{z}^1), \dots, \mathsf{vec}(\bar{z}^m)] \in \mathbb{R}^{(C \times T) \times m}, \\ \mathsf{circ}(\bar{Z}) &= [\mathsf{circ}(\bar{z}^1), \dots, \mathsf{circ}(\bar{z}^m)] \in \mathbb{R}^{(C \times T) \times (T \times m)}. \end{aligned} \tag{31}$$

Then, we define the *shift invariant coding rate reduction* for $\bar{Z} \in \mathbb{R}^{C \times T \times m}$ as

$$\begin{aligned} \Delta R_{\mathsf{circ}}(\bar{Z}, \Pi) &\doteq \frac{1}{T} \Delta R(\mathsf{circ}(\bar{Z}), \bar{\Pi}) \\ &= \frac{1}{2T} \log \det \left( I + \alpha \cdot \mathsf{circ}(\bar{Z}) \cdot \mathsf{circ}(\bar{Z})^* \right) - \sum_{j=1}^k \frac{\gamma_j}{2T} \log \det \left( I + \alpha_j \cdot \mathsf{circ}(\bar{Z}) \cdot \bar{\Pi}^j \cdot \mathsf{circ}(\bar{Z})^* \right), \end{aligned} \tag{32}$$

where $\alpha = \frac{CT}{mT\epsilon^2} = \frac{C}{m\epsilon^2}$, $\alpha_j = \frac{CT}{\mathsf{tr}(\Pi^j)T\epsilon^2} = \frac{C}{\mathsf{tr}(\Pi^j)\epsilon^2}$, $\gamma_j = \frac{\mathsf{tr}(\Pi^j)}{m}$, and $\bar{\Pi}^j$ is augmented membership matrix in an obvious way. Note that we introduce the normalization factor $T$ in (32) because the circulant matrix $\mathsf{circ}(\bar{Z})$ contains $T$ (shifted) copies of each signal.

By applying (4) and (5), we obtain the derivative of $\Delta R_{\mathsf{circ}}(\bar{Z}, \Pi)$ as

$$\begin{aligned} \frac{1}{2T} \frac{\partial \log \det \left( I + \alpha \mathsf{circ}(\bar{Z}) \mathsf{circ}(\bar{Z})^* \right)}{\partial \mathsf{vec}(\bar{Z})} &= \frac{1}{2T} \frac{\partial \log \det \left( I + \alpha \mathsf{circ}(\bar{Z}) \mathsf{circ}(\bar{Z})^* \right)}{\partial \mathsf{circ}(\bar{Z})} \frac{\partial \mathsf{circ}(\bar{Z})}{\partial \mathsf{vec}(\bar{Z})} \\ &= \underbrace{\alpha \left( I + \alpha \mathsf{circ}(\bar{Z}) \mathsf{circ}(\bar{Z})^* \right)^{-1}}_{\bar{E} \, \in \mathbb{R}^{(C \times T) \times (C \times T)}} \mathsf{vec}(\bar{Z}), \end{aligned} \tag{33}$$

---

[16]Notice that in the main paper, for simplicity, we have used $n$ to indicate both the 1D "temporal" or 2D "spatial" dimension of a signal, just to be consistent with the vector case, which corresponds to $T$ here. All notation should be clear within the context.

$$\frac{\gamma_j}{2T} \frac{\partial \log \det \left( \boldsymbol{I} + \alpha_j \mathsf{circ}(\bar{\boldsymbol{Z}}) \boldsymbol{\Pi}^j \mathsf{circ}(\bar{\boldsymbol{Z}})^* \right)}{\partial \mathsf{vec}(\bar{\boldsymbol{Z}})} = \gamma_j \underbrace{\alpha_j \left( \boldsymbol{I} + \alpha_j \mathsf{circ}(\bar{\boldsymbol{Z}}) \boldsymbol{\Pi}^j \mathsf{circ}(\bar{\boldsymbol{Z}})^* \right)^{-1}}_{\bar{\boldsymbol{C}}^j \ \in \mathbb{R}^{(C \times T) \times (C \times T)}} \mathsf{vec}(\bar{\boldsymbol{Z}}) \boldsymbol{\Pi}^j .$$

(34)

In the following, we show that $\bar{\boldsymbol{E}} \cdot \mathsf{vec}(\bar{\boldsymbol{z}})$ represents a multi-channel circular convolution. Note that

$$\bar{\boldsymbol{E}} = \alpha \begin{bmatrix} \boldsymbol{I} + \alpha \sum_{i=1}^m \mathsf{circ}(\boldsymbol{z}^i[1]) \mathsf{circ}(\boldsymbol{z}^i[1])^* & \cdots & \sum_{i=1}^m \mathsf{circ}(\boldsymbol{z}^i[1]) \mathsf{circ}(\boldsymbol{z}^i[C])^* \\ \vdots & \ddots & \vdots \\ \sum_{i=1}^m \mathsf{circ}(\boldsymbol{z}^i[C]) \mathsf{circ}(\boldsymbol{z}^i[1])^* & \cdots & \boldsymbol{I} + \sum_{i=1}^m \alpha \mathsf{circ}(\boldsymbol{z}^i[C]) \mathsf{circ}(\boldsymbol{z}^i[C])^* \end{bmatrix}^{-1} .$$

(35)

By using Fact 2, the matrix in the inverse above is a *block circulant matrix*, i.e., a block matrix where each block is a circulant matrix. A useful fact about the inverse of such a matrix is the following.

**Fact 3 (Inverse of block circulant matrices)** *The inverse of a block circulant matrix is a block circulant matrix (with respect to the same block partition).*

The main result of this subsection is the following.

**Proposition B.2 (Convolution structures of $\bar{\boldsymbol{E}}$ and $\bar{\boldsymbol{C}}^j$)** *Given a collection of multi-channel 1D signals $\{\bar{\boldsymbol{z}}^i \in \mathbb{R}^{C \times T}\}_{i=1}^m$, the matrix $\bar{\boldsymbol{E}}$ is a block circulant matrix, i.e.,*

$$\bar{\boldsymbol{E}} \doteq \begin{bmatrix} \bar{\boldsymbol{E}}_{1,1} & \cdots & \bar{\boldsymbol{E}}_{1,C} \\ \vdots & \ddots & \vdots \\ \bar{\boldsymbol{E}}_{C,1} & \cdots & \bar{\boldsymbol{E}}_{C,C} \end{bmatrix} ,$$

(36)

*where each $\bar{\boldsymbol{E}}_{c,c'} \in \mathbb{R}^{T \times T}$ is a circulant matrix. Moreover, $\bar{\boldsymbol{E}}$ represents a multi-channel circular convolution, i.e., for any multi-channel signal $\bar{\boldsymbol{z}} \in \mathbb{R}^{C \times T}$ we have*

$$\bar{\boldsymbol{E}} \cdot \mathsf{vec}(\bar{\boldsymbol{z}}) = \mathsf{vec}(\bar{\boldsymbol{e}} \circledast \bar{\boldsymbol{z}}).$$

*In above, $\bar{\boldsymbol{e}} \in \mathbb{R}^{C \times C \times T}$ is a multi-channel convolutional kernel with $\bar{\boldsymbol{e}}[c, c'] \in \mathbb{R}^T$ being the first column vector of $\bar{\boldsymbol{E}}_{c,c'}$, and $\bar{\boldsymbol{e}} \circledast \bar{\boldsymbol{z}} \in \mathbb{R}^{C \times T}$ is the multi-channel circular convolution (with "$\circledast$" overloading the notation from Eq. (26)) defined as*

$$(\bar{\boldsymbol{e}} \circledast \bar{\boldsymbol{z}})[c] = \sum_{c'=1}^C \bar{\boldsymbol{e}}[c, c'] \circledast \bar{\boldsymbol{z}}[c'], \quad \forall c = 1, \dots, C.$$

(37)

*Similarly, the matrices $\bar{\boldsymbol{C}}^j$ associated with any subsets of $\bar{\boldsymbol{Z}}$ are also multi-channel circular convolutions.*

Note that the calculation of $\bar{\boldsymbol{E}}$ in (35) requires inverting a matrix of size $(C \times T) \times (C \times T)$. In the following, we show that this computation can be accelerated by working in the frequency domain.

### B.3 FAST COMPUTATION IN SPECTRAL DOMAIN

**Circulant matrix and Discrete Fourier Transform.** A remarkable property of circulant matrices is that *they all share the same set of eigenvectors that form a unitary matrix*. We define the matrix:

$$\boldsymbol{F}_T \doteq \frac{1}{\sqrt{T}} \begin{bmatrix} \omega_T^0 & \omega_T^0 & \cdots & \omega_T^0 & \omega_T^0 \\ \omega_T^0 & \omega_T^1 & \cdots & \omega_T^{T-2} & \omega_T^{T-1} \\ \vdots & \vdots & \ddots & \vdots & \vdots \\ \omega_T^0 & \omega_T^{T-2} & \cdots & \omega_T^{(T-2)^2} & \omega_T^{(T-2)(T-1)} \\ \omega_T^0 & \omega_T^{T-1} & \cdots & \omega_T^{(T-2)(T-1)} & \omega_T^{(T-1)^2} \end{bmatrix} \in \mathbb{C}^{T \times T},$$

(38)

where $\omega_T \doteq \exp(-\frac{2\pi\sqrt{-1}}{T})$ is the roots of unit (as $\omega_T^T = 1$). The matrix $\boldsymbol{F}_T$ is a unitary matrix: $\boldsymbol{F}_T \boldsymbol{F}_T^* = \boldsymbol{I}$ and is the well known *Vandermonde matrix*. Multiplying a vector with $\boldsymbol{F}_T$ is known as the *discrete Fourier transform* (DFT). Be aware that the conventional DFT matrix differs from our definition of $\boldsymbol{F}_T$ here by a scale: it does not have the $\frac{1}{\sqrt{T}}$ in front. Here for simplicity, we scale it so that $\boldsymbol{F}_T$ is a unitary matrix and its inverse is simply its conjugate transpose $\boldsymbol{F}_T^*$, columns of which represent the eigenvectors of a circulant matrix (Abidi et al., 2016).

**Fact 4 (DFT as matrix-vector multiplication)** *The DFT of a vector $\boldsymbol{z} \in \mathbb{R}^T$ can be computed as*

$$\mathrm{DFT}(\boldsymbol{z}) \doteq \boldsymbol{F}_T \cdot \boldsymbol{z} \quad \in \mathbb{C}^T, \tag{39}$$

*where*

$$\mathrm{DFT}(\boldsymbol{z})(p) = \frac{1}{\sqrt{T}} \sum_{t=0}^{T-1} z(t) \cdot \omega_T^{p \cdot t}, \quad \forall p = 0, 1, \ldots, T - 1. \tag{40}$$

*The Inverse Discrete Fourier Transform (IDFT) of a signal $\boldsymbol{v} \in \mathbb{C}^T$ can be computed as*

$$\mathrm{IDFT}(\boldsymbol{v}) \doteq \boldsymbol{F}_T^* \cdot \boldsymbol{v} \quad \in \mathbb{C}^T \tag{41}$$

*where*

$$\mathrm{IDFT}(\boldsymbol{v})(t) = \frac{1}{\sqrt{T}} \sum_{p=0}^{T-1} v(p) \cdot \omega_T^{-p \cdot t}, \quad \forall t = 0, 1, \ldots, T - 1. \tag{42}$$

Regarding the relationship between a circulant matrix (convolution) and discrete Fourier transform, we have:

**Fact 5** *An $n \times n$ matrix $\boldsymbol{M} \in \mathbb{C}^{n \times n}$ is a circulant matrix if and only if it is diagonalizable by the unitary matrix $\boldsymbol{F}_n$:*

$$\boldsymbol{F}_n \boldsymbol{M} \boldsymbol{F}_n^* = \boldsymbol{D} \quad or \quad \boldsymbol{M} = \boldsymbol{F}_n^* \boldsymbol{D} \boldsymbol{F}_n, \tag{43}$$

*where $\boldsymbol{D}$ is a diagonal matrix of eigenvalues.*

**Fact 6 (DFT are eigenvalues of the circulant matrix)** *Given a vector $\boldsymbol{z} \in \mathbb{C}^T$, we have*

$$\boldsymbol{F}_T \cdot \mathrm{circ}(\boldsymbol{z}) \cdot \boldsymbol{F}_T^* = \mathrm{diag}(\mathrm{DFT}(\boldsymbol{z})) \quad or \quad \mathrm{circ}(\boldsymbol{z}) = \boldsymbol{F}_T^* \cdot \mathrm{diag}(\mathrm{DFT}(\boldsymbol{z})) \cdot \boldsymbol{F}_T. \tag{44}$$

*That is, the eigenvalues of the circulant matrix associated with a vector are given by its DFT.*

**Fact 7 (Parseval's theorem)** *Given any $\boldsymbol{z} \in \mathbb{C}^T$, we have $\|\boldsymbol{z}\|_2 = \|\mathrm{DFT}(\boldsymbol{z})\|_2$. More precisely,*

$$\sum_{t=0}^{T-1} |\boldsymbol{z}[t]|^2 = \sum_{p=0}^{T-1} |\mathrm{DFT}(\boldsymbol{z})[p]|^2. \tag{45}$$

This property allows us to easily "normalize" features after each layer onto the sphere $\mathbb{S}^{n-1}$ directly in the spectral domain (see Eq. (10) and (62)).

**Circulant matrix and Discrete Fourier Transform for multi-channel signals.** We now consider multi-channel 1D signals $\bar{\boldsymbol{z}} \in \mathbb{R}^{C \times T}$. Let $\mathrm{DFT}(\bar{\boldsymbol{z}}) \in \mathbb{C}^{C \times T}$ be a matrix where the $c$-th row is the DFT of the corresponding signal $\boldsymbol{z}[c]$, i.e.,

$$\mathrm{DFT}(\bar{\boldsymbol{z}}) \doteq \begin{bmatrix} \mathrm{DFT}(\boldsymbol{z}[1])^* \\ \vdots \\ \mathrm{DFT}(\boldsymbol{z}[C])^* \end{bmatrix} \quad \in \mathbb{C}^{C \times T}. \tag{46}$$

Similar to the notation in (27), we denote

$$\mathrm{DFT}(\bar{\boldsymbol{z}}) = \begin{bmatrix} \mathrm{DFT}(\bar{\boldsymbol{z}})[1]^* \\ \vdots \\ \mathrm{DFT}(\bar{\boldsymbol{z}})[C]^* \end{bmatrix} = [\mathrm{DFT}(\bar{\boldsymbol{z}})(0), \mathrm{DFT}(\bar{\boldsymbol{z}})(1), \ldots, \mathrm{DFT}(\bar{\boldsymbol{z}})(T-1)]$$
$$= \{\mathrm{DFT}(\bar{\boldsymbol{z}})[c](t)\}_{c=1,t=0}^{c=C,t=T-1}. \tag{47}$$

As such, we have $\mathrm{DFT}(\boldsymbol{z}[c]) = \mathrm{DFT}(\bar{\boldsymbol{z}})[c]$.

By using Fact 6, $\mathrm{circ}(\bar{\boldsymbol{z}})$ and $\mathrm{DFT}(\bar{\boldsymbol{z}})$ are related as follows:

$$\mathrm{circ}(\bar{\boldsymbol{z}}) = \begin{bmatrix} \boldsymbol{F}_T^* \cdot \mathrm{diag}(\mathrm{DFT}(\boldsymbol{z}[1])) \cdot \boldsymbol{F}_T \\ \vdots \\ \boldsymbol{F}_T^* \cdot \mathrm{diag}(\mathrm{DFT}(\boldsymbol{z}[C])) \cdot \boldsymbol{F}_T \end{bmatrix} = \begin{bmatrix} \boldsymbol{F}_T^* & \cdots & \boldsymbol{0} \\ \vdots & \ddots & \vdots \\ \boldsymbol{0} & \cdots & \boldsymbol{F}_T^* \end{bmatrix} \begin{bmatrix} \mathrm{diag}(\mathrm{DFT}(\boldsymbol{z}[1])) \\ \vdots \\ \mathrm{diag}(\mathrm{DFT}(\boldsymbol{z}[C])) \end{bmatrix} \cdot \boldsymbol{F}_T. \tag{48}$$

We now explain how this relationship can be leveraged to produce a fast computation of $\bar{\boldsymbol{E}}$ defined in (33). First, there exists a permutation matrix $\mathbf{P}$ such that

$$\begin{bmatrix} \operatorname{diag}(\operatorname{DFT}(\boldsymbol{z}[1])) \\ \operatorname{diag}(\operatorname{DFT}(\boldsymbol{z}[2])) \\ \vdots \\ \operatorname{diag}(\operatorname{DFT}(\boldsymbol{z}[C])) \end{bmatrix} = \mathbf{P} \cdot \begin{bmatrix} \operatorname{DFT}(\bar{z})(0) & \mathbf{0} & \cdots & 0 \\ 0 & \operatorname{DFT}(\bar{z})(1) & \cdots & 0 \\ \vdots & \vdots & \ddots & \vdots \\ 0 & 0 & \cdots & \operatorname{DFT}(\bar{z})(T-1) \end{bmatrix}. \tag{49}$$

Combining (48) and (49), we have

$$\operatorname{circ}(\bar{z}) \cdot \operatorname{circ}(\bar{z})^* = \begin{bmatrix} \boldsymbol{F}_T^* & \cdots & \mathbf{0} \\ \vdots & \ddots & \vdots \\ \mathbf{0} & \cdots & \boldsymbol{F}_T^* \end{bmatrix} \cdot \mathbf{P} \cdot \boldsymbol{D}(\bar{z}) \cdot \mathbf{P}^* \cdot \begin{bmatrix} \boldsymbol{F}_T & \cdots & \mathbf{0} \\ \vdots & \ddots & \vdots \\ \mathbf{0} & \cdots & \boldsymbol{F}_T \end{bmatrix}, \tag{50}$$

where

$$\boldsymbol{D}(\bar{z}) \doteq \begin{bmatrix} \operatorname{DFT}(\bar{z})(0) \cdot \operatorname{DFT}(\bar{z})(0)^* & \cdots & & \mathbf{0} \\ \vdots & \ddots & & \vdots \\ \mathbf{0} & \cdots & \operatorname{DFT}(\bar{z})(T-1) \cdot \operatorname{DFT}(\bar{z})(T-1)^* \end{bmatrix}. \tag{51}$$

Now, consider a collection of $m$ multi-channel 1D signals $\bar{\boldsymbol{Z}} \in \mathbb{R}^{C \times T \times m}$. Similar to the notation in (30), we denote

$$\begin{aligned} \operatorname{DFT}(\bar{\boldsymbol{Z}})[c] &= [\operatorname{DFT}(\bar{z}^1)[c], \ldots, \operatorname{DFT}(\bar{z}^m)[c]] \in \mathbb{R}^{T \times m}, \\ \operatorname{DFT}(\bar{\boldsymbol{Z}})(p) &= [\operatorname{DFT}(\bar{z}^1)(p), \ldots, \operatorname{DFT}(\bar{z}^m)(p)] \in \mathbb{R}^{C \times m}. \end{aligned} \tag{52}$$

By using (50), we have

$$\bar{\boldsymbol{E}} = \begin{bmatrix} \boldsymbol{F}_T^* & \cdots & \mathbf{0} \\ \vdots & \ddots & \vdots \\ \mathbf{0} & \cdots & \boldsymbol{F}_T^* \end{bmatrix} \cdot \mathbf{P} \cdot \alpha \cdot \left[ \boldsymbol{I} + \alpha \cdot \sum_{i=1}^m \boldsymbol{D}(\bar{z}^i) \right]^{-1} \cdot \mathbf{P}^* \cdot \begin{bmatrix} \boldsymbol{F}_T & \cdots & \mathbf{0} \\ \vdots & \ddots & \vdots \\ \mathbf{0} & \cdots & \boldsymbol{F}_T \end{bmatrix}. \tag{53}$$

Note that $\alpha \cdot \left[ \boldsymbol{I} + \alpha \cdot \sum_{i=1}^m \boldsymbol{D}(\bar{z}^i) \right]^{-1}$ is equal to

$$\begin{aligned} &\alpha \begin{bmatrix} \boldsymbol{I} + \alpha \operatorname{DFT}(\bar{\boldsymbol{Z}})(0) \cdot \operatorname{DFT}(\bar{\boldsymbol{Z}}^i)(0)^* & \cdots & \mathbf{0} \\ \vdots & \ddots & \vdots \\ \mathbf{0} & \cdots & \boldsymbol{I} + \alpha \operatorname{DFT}(\bar{\boldsymbol{Z}})(T-1) \cdot \operatorname{DFT}(\bar{\boldsymbol{Z}})(T-1)^* \end{bmatrix}^{-1} \\ &= \begin{bmatrix} \alpha \left( \boldsymbol{I} + \alpha \operatorname{DFT}(\bar{\boldsymbol{Z}})(0) \cdot \operatorname{DFT}(\bar{\boldsymbol{Z}})(0)^* \right)^{-1} & \cdots & \mathbf{0} \\ \vdots & \ddots & \vdots \\ \mathbf{0} & \cdots & \alpha \left( \boldsymbol{I} + \alpha \operatorname{DFT}(\bar{\boldsymbol{Z}})(T-1) \cdot \operatorname{DFT}(\bar{\boldsymbol{Z}})(T-1)^* \right)^{-1} \end{bmatrix}. \end{aligned} \tag{54}$$

Therefore, the calculation of $\bar{\boldsymbol{E}}$ only requires inverting $T$ matrices of size $C \times C$. This motivates us to construct the ReduNet in the spectral domain for the purpose of accelerating the computation, as we explain next.

**Shift-invariant ReduNet in the Spectral Domain.** Motivated by the result in (54), we introduce the notations $\bar{\mathcal{E}}(p) \in \mathbb{R}^{C \times C \times T}$ and $\bar{\mathcal{C}}^j(p) \in \mathbb{R}^{C \times C \times T}$ given by

$$\begin{aligned} \bar{\mathcal{E}}(p) &\doteq \alpha \cdot \left[ \boldsymbol{I} + \alpha \cdot \operatorname{DFT}(\bar{\boldsymbol{Z}})(p) \cdot \operatorname{DFT}(\bar{\boldsymbol{Z}})(p)^* \right]^{-1} &\in \mathbb{C}^{C \times C}, \tag{55} \\ \bar{\mathcal{C}}^j(p) &\doteq \alpha_j \cdot \left[ \boldsymbol{I} + \alpha_j \cdot \operatorname{DFT}(\bar{\boldsymbol{Z}})(p) \cdot \boldsymbol{\Pi}_j \cdot \operatorname{DFT}(\bar{\boldsymbol{Z}})(p)^* \right]^{-1} &\in \mathbb{C}^{C \times C}. \tag{56} \end{aligned}$$

In above, $\bar{\mathcal{E}}(p)$ (resp., $\bar{\mathcal{C}}^j(p)$) is the $p$-th slice of $\bar{\mathcal{E}}$ (resp., $\bar{\mathcal{C}}^j$) on the last dimension. Then, the gradient of $\Delta R_{\operatorname{circ}}(\bar{\boldsymbol{Z}}, \boldsymbol{\Pi})$ with respect to $\bar{\boldsymbol{Z}}$ can be calculated by the following result.

**Theorem B.3 (Computing multi-channel convolutions $\bar{\boldsymbol{E}}$ and $\bar{\boldsymbol{C}}^j$)** *Let* $\bar{\boldsymbol{U}} \in \mathbb{C}^{C \times T \times m}$ *and* $\bar{\boldsymbol{W}}^j \in \mathbb{C}^{C \times T \times m}, j = 1, \ldots, k$ *be given by*

$$\begin{aligned} \bar{\boldsymbol{U}}(p) &\doteq \bar{\mathcal{E}}(p) \cdot \operatorname{DFT}(\bar{\boldsymbol{Z}})(p), \tag{57} \\ \bar{\boldsymbol{W}}^j(p) &\doteq \bar{\mathcal{C}}^j(p) \cdot \operatorname{DFT}(\bar{\boldsymbol{Z}})(p), \quad j = 1, \ldots, k, \tag{58} \end{aligned}$$

*for each $p \in \{0, \ldots, T-1\}$. Then, we have*

$$\frac{1}{2T} \frac{\partial \log \det(\boldsymbol{I} + \alpha \cdot \text{circ}(\bar{\boldsymbol{Z}})\text{circ}(\bar{\boldsymbol{Z}})^*)}{\partial \bar{\boldsymbol{Z}}} = \text{IDFT}(\bar{\boldsymbol{U}}), \tag{59}$$

$$\frac{\gamma_j}{2T} \frac{\partial \log \det(\boldsymbol{I} + \alpha_j \cdot \text{circ}(\bar{\boldsymbol{Z}})\bar{\boldsymbol{\Pi}}^j \text{circ}(\bar{\boldsymbol{Z}})^*)}{\partial \bar{\boldsymbol{Z}}} = \gamma_j \cdot \text{IDFT}(\bar{\boldsymbol{W}}^j \boldsymbol{\Pi}^j). \tag{60}$$

By this result, the gradient ascent update in (3) (when applied to $\Delta R_{\text{circ}}(\bar{\boldsymbol{Z}}, \boldsymbol{\Pi})$) can be equivalently expressed as an update in frequency domain on $\bar{V}_\ell \doteq \text{DFT}(\bar{\boldsymbol{Z}}_\ell)$ as

$$\bar{V}_{\ell+1}(p) \propto \bar{V}_\ell(p) + \eta \left( \bar{\mathcal{E}}_\ell(p) \cdot \bar{V}_\ell(p) - \sum_{j=1}^{k} \gamma_j \bar{\mathcal{C}}_\ell^j(p) \cdot \bar{V}_\ell(p)\Pi^j \right), \quad p = 0, \ldots, T-1. \tag{61}$$

Similarly, the gradient-guided feature map increment in (10) can be equivalently expressed as an update in frequency domain on $\bar{\boldsymbol{v}}_\ell \doteq \text{DFT}(\bar{\boldsymbol{z}}_\ell)$ as

$$\bar{\boldsymbol{v}}_{\ell+1}(p) \propto \bar{\boldsymbol{v}}_\ell(p) + \eta \cdot \bar{\mathcal{E}}_\ell(p)\bar{\boldsymbol{v}}_\ell(p) - \eta \cdot \boldsymbol{\sigma}\left([\bar{\mathcal{C}}_\ell^1(p)\bar{\boldsymbol{v}}_\ell(p), \ldots, \bar{\mathcal{C}}_\ell^k(p)\bar{\boldsymbol{v}}_\ell(p)]\right), \quad p = 0, \ldots, T-1, \tag{62}$$

subject to the constraint that $\|\bar{\boldsymbol{v}}_{\ell+1}\|_F = \|\bar{\boldsymbol{z}}_{\ell+1}\|_F = 1$ (the first equality follows from Fact 7).

We summarize the training, or construction to be more precise, of ReduNet in the spectral domain in Algorithm 1.

**Proof:** [Proof to Theorem (B.3)]

From (4), (53) and (48), we have

$$\frac{1}{2} \frac{\partial \log \det \left( \boldsymbol{I} + \alpha \text{circ}(\bar{\boldsymbol{Z}})\text{circ}(\bar{\boldsymbol{Z}})^* \right)}{\partial \text{circ}(\bar{\boldsymbol{z}}^i)} = \bar{\boldsymbol{E}}\text{circ}(\bar{\boldsymbol{z}}^i) = \bar{\boldsymbol{E}} \begin{bmatrix} \boldsymbol{F}_T^* & \cdots & \boldsymbol{0} \\ \vdots & \ddots & \vdots \\ \boldsymbol{0} & \cdots & \boldsymbol{F}_T^* \end{bmatrix} \begin{bmatrix} \text{diag}(\text{DFT}(\boldsymbol{z}^i[1])) \\ \vdots \\ \text{diag}(\text{DFT}(\boldsymbol{z}^i[C])) \end{bmatrix} \boldsymbol{F}_T \tag{63}$$

$$= \begin{bmatrix} \boldsymbol{F}_T^* & \cdots & \boldsymbol{0} \\ \vdots & \ddots & \vdots \\ \boldsymbol{0} & \cdots & \boldsymbol{F}_T^* \end{bmatrix} \cdot \boldsymbol{P} \cdot \alpha \cdot \left[ \boldsymbol{I} + \alpha \cdot \sum_i \boldsymbol{D}(\bar{\boldsymbol{z}}^i) \right]^{-1} \cdot \begin{bmatrix} \text{DFT}(\bar{\boldsymbol{z}}^i)(0) & \cdots & \boldsymbol{0} \\ \vdots & \ddots & \vdots \\ \boldsymbol{0} & \cdots & \text{DFT}(\bar{\boldsymbol{z}}^i)(T-1) \end{bmatrix} \cdot \boldsymbol{F}_T \tag{64}$$

$$= \begin{bmatrix} \boldsymbol{F}_T^* & \cdots & \boldsymbol{0} \\ \vdots & \ddots & \vdots \\ \boldsymbol{0} & \cdots & \boldsymbol{F}_T^* \end{bmatrix} \cdot \boldsymbol{P} \cdot \begin{bmatrix} \bar{\mathcal{E}}(0) \cdot \text{DFT}(\bar{\boldsymbol{z}}^i)(0) & \cdots & \boldsymbol{0} \\ \vdots & \ddots & \vdots \\ \boldsymbol{0} & \cdots & \bar{\mathcal{E}}(T-1) \cdot \text{DFT}(\bar{\boldsymbol{z}}^i)(T-1) \end{bmatrix} \cdot \boldsymbol{F}_T \tag{65}$$

$$= \begin{bmatrix} \boldsymbol{F}_T^* & \cdots & \boldsymbol{0} \\ \vdots & \ddots & \vdots \\ \boldsymbol{0} & \cdots & \boldsymbol{F}_T^* \end{bmatrix} \cdot \boldsymbol{P} \cdot \begin{bmatrix} \bar{\boldsymbol{u}}^i(0) & \cdots & \boldsymbol{0} \\ \vdots & \ddots & \vdots \\ \boldsymbol{0} & \cdots & \bar{\boldsymbol{u}}^i(T-1) \end{bmatrix} \cdot \boldsymbol{F}_T = \begin{bmatrix} \boldsymbol{F}_T^* & \cdots & \boldsymbol{0} \\ \vdots & \ddots & \vdots \\ \boldsymbol{0} & \cdots & \boldsymbol{F}_T^* \end{bmatrix} \cdot \begin{bmatrix} \text{diag}(\bar{\boldsymbol{u}}^i[1]) \\ \vdots \\ \text{diag}(\bar{\boldsymbol{u}}^i[C]) \end{bmatrix} \cdot \boldsymbol{F}_T \tag{66}$$

$$= \text{circ}(\text{IDFT}(\bar{\boldsymbol{u}}^i)). \tag{67}$$

Therefore, we have

$$\frac{1}{2} \frac{\partial \log \det \left( \boldsymbol{I} + \alpha \cdot \text{circ}(\bar{\boldsymbol{Z}}) \cdot \text{circ}(\bar{\boldsymbol{Z}})^* \right)}{\partial \bar{\boldsymbol{z}}^i} = \frac{1}{2} \frac{\partial \log \det \left( \boldsymbol{I} + \alpha \cdot \text{circ}(\bar{\boldsymbol{Z}}) \cdot \text{circ}(\bar{\boldsymbol{Z}})^* \right)}{\partial \text{circ}(\bar{\boldsymbol{z}}^i)} \cdot \frac{\partial \text{circ}(\bar{\boldsymbol{z}}^i)}{\partial \bar{\boldsymbol{z}}^i}$$
$$= T \cdot \text{IDFT}(\bar{\boldsymbol{u}}^i). \tag{68}$$

By collecting the results for all $i$, we have

$$\frac{\partial \frac{1}{2T} \log \det \left( \boldsymbol{I} + \alpha \cdot \text{circ}(\bar{\boldsymbol{Z}}) \cdot \text{circ}(\bar{\boldsymbol{Z}})^* \right)}{\partial \bar{\boldsymbol{Z}}} = \text{IDFT}(\bar{\boldsymbol{U}}). \tag{69}$$

In a similar fashion, we get

$$\frac{\partial \frac{\gamma_j}{2T} \log \det \left( \boldsymbol{I} + \alpha_j \cdot \text{circ}(\bar{\boldsymbol{Z}}) \cdot \bar{\boldsymbol{\Pi}}^j \cdot \text{circ}(\bar{\boldsymbol{Z}})^* \right)}{\partial \bar{\boldsymbol{Z}}} = \gamma_j \cdot \text{IDFT}(\bar{\boldsymbol{W}}^j \cdot \boldsymbol{\Pi}^j). \tag{70}$$

$\square$

---

**Algorithm 1 Training Algorithm** (1D Signal, Shift Invariance, Spectral Domain)

---

**Input:** $\bar{Z} \in \mathbb{R}^{C \times T \times m}$, $\mathbf{\Pi}$, $\epsilon > 0$, $\lambda$, and a learning rate $\eta$.

1: Set $\alpha = \frac{C}{m\epsilon^2}$, $\{\alpha_j = \frac{C}{\text{tr}(\mathbf{\Pi}^j)\epsilon^2}\}_{j=1}^k$, $\{\gamma_j = \frac{\text{tr}(\mathbf{\Pi}^j)}{m}\}_{j=1}^k$.

2: Set $\bar{V}_0 = \{\bar{v}_0^i(p) \in \mathbb{C}^C\}_{p=0,i=1}^{T-1,m} \doteq \text{DFT}(\bar{Z}) \in \mathbb{C}^{C \times T \times m}$.

3: **for** $\ell = 1, 2, \ldots, L$ **do**

4:      *# Step 1: Compute $\mathcal{E}$ and $\mathcal{C}$.*

5:      **for** $p = 0, 1, \ldots, T-1$ **do**

6:          Compute $\bar{\mathcal{E}}_\ell(p) \in \mathbb{C}^{C \times C}$ and $\{\bar{\mathcal{C}}_\ell^j(p) \in \mathbb{C}^{C \times C}\}_{j=1}^k$ as
$$\bar{\mathcal{E}}_\ell(p) \doteq \alpha \cdot \left[ \mathbf{I} + \alpha \cdot \bar{V}_{\ell-1}(p) \cdot \bar{V}_{\ell-1}(p)^* \right]^{-1},$$
$$\bar{\mathcal{C}}_\ell^j(p) \doteq \alpha_j \cdot \left[ \mathbf{I} + \alpha_j \cdot \bar{V}_{\ell-1}(p) \cdot \mathbf{\Pi}^j \cdot \bar{V}_{\ell-1}(p)^* \right]^{-1};$$

7:      **end for**

8:      *# Step 2: Update $\bar{v}^i$ for each $i$.*

9:      **for** $i = 1, \ldots, m$ **do**

10:        *# Compute projection at each frequency $p$.*

11:        **for** $p = 0, 1, \ldots, T-1$ **do**

12:          Compute $\{\bar{p}_\ell^{ij}(p) \doteq \bar{\mathcal{C}}_\ell^j(p) \cdot \bar{v}_\ell^i(p) \in \mathbb{C}^{C \times 1}\}_{j=1}^k$;

13:        **end for**

14:        *# Compute overall projection by aggregating over frequency $p$.*

15:        Let $\{\bar{\mathbf{P}}_\ell^{ij} = [\bar{p}_\ell^{ij}(0), \ldots, \bar{p}_\ell^{ij}(T-1)] \in \mathbb{C}^{C \times T}\}_{j=1}^k$;

16:        *# Compute soft assignment from projection.*

17:        Compute $\left\{ \hat{\pi}_\ell^{ij} = \frac{\exp(-\lambda \|\bar{\mathbf{P}}_\ell^{ij}\|_F)}{\sum_{j=1}^k \exp(-\lambda \|\bar{\mathbf{P}}_\ell^{ij}\|_F)} \right\}_{j=1}^k$;

18:        *# Compute update at each frequency $p$.*

19:        **for** $p = 0, 1, \ldots, T-1$ **do**

20:          $\bar{v}_\ell^i(p) = \bar{v}_{\ell-1}^i(p) + \eta \left( \bar{\mathcal{E}}_\ell(p)\bar{v}_\ell^i(p) - \sum_{j=1}^k \gamma_j \cdot \hat{\pi}_\ell^{ij} \cdot \bar{\mathcal{C}}_\ell^j(p) \cdot \bar{v}_\ell^i(p) \right)$;

21:        **end for**

22:        $\bar{v}_\ell^i = \bar{v}_\ell^i \, / \, \|\bar{v}_\ell^i\|_F$;

23:      **end for**

24:      Set $\bar{Z}_\ell = \text{IDFT}(\bar{V}_\ell)$ as the feature at the $\ell$-th layer;

25:      *# Evaluate the objective value.*

26:      $\frac{1}{2T} \sum_{p=0}^{T-1} \left( \log\det[\mathbf{I} + \alpha \bar{V}_\ell(p) \cdot \bar{V}_\ell(p)^*] - \frac{\text{tr}(\mathbf{\Pi}^j)}{m} \log\det[\mathbf{I} + \alpha_j \bar{V}_\ell(p) \cdot \mathbf{\Pi}^j \cdot \bar{V}_\ell(p)^*] \right)$;

27: **end for**

**Output:** features $\bar{Z}_L$, the learned filters $\{\bar{E}_\ell(p)\}_{\ell,p}$ and $\{\bar{\mathcal{C}}_\ell^j(p)\}_{j,\ell,p}$.

---

## C  2D CIRCULAR TRANSLATION INVARIANCE

To a large degree, both conceptually and technically, the 2D case is very similar to the 1D case that we have studied carefully in the previous Appendix B. For the sake of consistency and completeness, we here gives a brief account.

### C.1  DOUBLY BLOCK CIRCULANT MATRIX

In this section, we consider $z$ as a 2D signal such as an image, and use $H$ and $W$ to denote its "height" and "width", respectively. It will be convenient to work with both a matrix representation

$$z = \begin{bmatrix} z(0,0) & z(0,1) & \cdots & z(0,W-1) \\ z(1,0) & z(1,1) & \cdots & z(1,W-1) \\ \vdots & \vdots & \ddots & \vdots \\ z(H-1,0) & z(H-1,1) & \cdots & z(H-1,W-1) \end{bmatrix} \in \mathbb{R}^{H \times W}, \quad (71)$$

as well as a vector representation

$$\mathsf{vec}(z) \doteq \Big[ z(0,0), \ldots, z(0,W-1), z(1,0), \ldots, z(1,W-1), \ldots$$

$$\ldots, z(H-1,0), \ldots, z(H-1,W-1) \Big]^* \in \mathbb{R}^{(H \times W)}. \quad (72)$$

We represent the circular translated version of $z$ as $\mathsf{trans}_{p,q}(z) \in \mathbb{R}^{H \times W}$ by an amount of $p$ and $q$ on the vertical and horizontal directions, respectively. That is, we let

$$\mathsf{trans}_{p,q}(z)(h,w) \doteq z(h-p \bmod H, w-q \bmod W),$$
$$\forall (h,w) \in \{0,\ldots,H-1\} \times \{0,\ldots,W-1\}. \quad (73)$$

It is obvious that $\mathsf{trans}_{0,0}(z) = z$. Moreover, there is a total number of $H \times W$ distinct translations given by $\{\mathsf{trans}_{p,q}(z), (p,q) \in \{0,\ldots,H-1\} \times \{0,\ldots,W-1\}\}$. We may arrange the vector representations of them into a matrix and obtain

$$\mathsf{circ}(z) \doteq \Big[ \mathsf{vec}(\mathsf{trans}_{0,0}(z)), \ldots, \mathsf{vec}(\mathsf{trans}_{0,W-1}(z)),$$

$$\mathsf{vec}(\mathsf{trans}_{1,0}(z)), \ldots, \mathsf{vec}(\mathsf{trans}_{1,W-1}(z)),$$

$$\cdots,$$

$$\mathsf{vec}(\mathsf{trans}_{H-1,0}(z)), \ldots, \mathsf{vec}(\mathsf{trans}_{H-1,W-1}(z)) \Big] \in \mathbb{R}^{(H \times W) \times (H \times W)}. \quad (74)$$

The matrix $\mathsf{circ}(z)$ is known as the *doubly block circulant matrix* associated with $z$ (see, e.g., Abidi et al. (2016); Sedghi et al. (2018)).

We now consider a multi-channel 2D signal represented as a tensor $\bar{z} \in \mathbb{R}^{C \times H \times W}$, where $C$ is the number of channels. The $c$-th channel of $\bar{z}$ is represented as $\bar{z}[c] \in \mathbb{R}^{H \times W}$, and the $(h,w)$-th pixel is represented as $\bar{z}(h,w) \in \mathbb{R}^C$. To compute the coding rate reduction for a collection of such multi-channel 2D signals, we may flatten the tenor representation into a vector representation by concatenating the vector representation of each channel, i.e., we let

$$\mathsf{vec}(\bar{z}) = [\mathsf{vec}(\bar{z}[1])^*, \ldots, \mathsf{vec}(\bar{z}[C])^*]^* \in \mathbb{R}^{(C \times H \times W)} \quad (75)$$

Furthermore, to obtain shift invariance for coding rate reduction, we may generate a collection of translated versions of $\bar{z}$ (along two spatial dimensions). Stacking the vector representation for such translated copies as column vectors, we obtain

$$\mathsf{circ}(\bar{z}) \doteq \begin{bmatrix} \mathsf{circ}(\bar{z}[1]) \\ \vdots \\ \mathsf{circ}(\bar{z}[C]) \end{bmatrix} \in \mathbb{R}^{(C \times H \times W) \times (H \times W)}. \quad (76)$$

We can now define a *translation invariant coding rate reduction* for multi-channel 2D signals. Consider a collection of $m$ multi-channel 2D signals $\{\bar{z}^i \in \mathbb{R}^{C \times H \times W}\}_{i=1}^m$. Compactly representing the data by $\bar{Z} \in \mathbb{R}^{C \times H \times W \times m}$ where the $i$-th slice on the last dimension is $\bar{z}^i$, we denote

$$\mathsf{circ}(\bar{Z}) = [\mathsf{circ}(\bar{z}^1), \ldots, \mathsf{circ}(\bar{z}^m)] \in \mathbb{R}^{(C \times H \times W) \times (H \times W \times m)}. \quad (77)$$

Then, we define

$$
\Delta R_{\text{circ}}(\bar{\boldsymbol{Z}}, \boldsymbol{\Pi}) \doteq \frac{1}{HW} \Delta R(\text{circ}(\bar{\boldsymbol{Z}}), \bar{\boldsymbol{\Pi}}) = \frac{1}{2HW} \log \det \left( \boldsymbol{I} + \alpha \cdot \text{circ}(\bar{\boldsymbol{Z}}) \cdot \text{circ}(\bar{\boldsymbol{Z}})^* \right)
$$
$$
- \sum_{j=1}^{k} \frac{\gamma_j}{2HW} \log \det \left( \boldsymbol{I} + \alpha_j \cdot \text{circ}(\bar{\boldsymbol{Z}}) \cdot \bar{\boldsymbol{\Pi}}^j \cdot \text{circ}(\bar{\boldsymbol{Z}})^* \right), \quad (78)
$$

where $\alpha = \frac{CHW}{mHW\epsilon^2} = \frac{C}{m\epsilon^2}$, $\alpha_j = \frac{CHW}{\text{tr}(\boldsymbol{\Pi}^j)HW\epsilon^2} = \frac{C}{\text{tr}(\boldsymbol{\Pi}^j)\epsilon^2}$, $\gamma_j = \frac{\text{tr}(\boldsymbol{\Pi}^j)}{m}$, and $\bar{\boldsymbol{\Pi}}^j$ is augmented membership matrix in an obvious way.

By following an analogous argument as in the 1D case, one can show that ReduNet for multi-channel 2D signals naturally gives rise to the multi-channel 2D circulant convolution operations. We omit the details, and focus on the construction of ReduNet in the frequency domain.

## C.2 FAST COMPUTATION IN SPECTRAL DOMAIN

**Doubly block circulant matrix and 2D-DFT.** Similar to the case of circulant matrices for 1D signals, all doubly block circulant matrices share the same set of eigenvectors, and these eigenvectors form a unitary matrix given by

$$
\boldsymbol{F} \doteq \boldsymbol{F}_H \otimes \boldsymbol{F}_W \quad \in \mathbb{C}^{(H \times W) \times (H \times W)}, \quad (79)
$$

where $\otimes$ denotes the Kronecker product and $\boldsymbol{F}_H, \boldsymbol{F}_W$ are defined as in (38).

Analogous to Fact 4, $\boldsymbol{F}$ defines 2D-DFT as follows.

**Fact 8 (2D-DFT as matrix-vector multiplication)** *The 2D-DFT of a signal $\boldsymbol{z} \in \mathbb{R}^{H \times W}$ can be computed as*

$$
\text{vec}(\text{DFT}(\boldsymbol{z})) \doteq \boldsymbol{F} \cdot \text{vec}(\boldsymbol{z}) \quad \in \mathbb{C}^{(H \times W)}, \quad (80)
$$

*where*

$$
\text{DFT}(\boldsymbol{z})(p, q) = \frac{1}{\sqrt{H \cdot W}} \sum_{h=0}^{H-1} \sum_{w=0}^{W-1} \boldsymbol{z}(h, w) \cdot \omega_H^{p \cdot h} \omega_W^{q \cdot w},
$$
$$
\forall (p, q) \in \{0, \ldots, H-1\} \times \{0, \ldots, W-1\}. \quad (81)
$$

*The 2D-IDFT of a signal $\boldsymbol{v} \in \mathbb{C}^{H \times W}$ can be computed as*

$$
\text{vec}(\text{IDFT}(\boldsymbol{v})) \doteq \boldsymbol{F}_T^* \cdot \text{vec}(\boldsymbol{v}) \quad \in \mathbb{C}^{(H \times W)}, \quad (82)
$$

*where*

$$
\text{IDFT}(\boldsymbol{v})(h, w) = \frac{1}{\sqrt{H \cdot W}} \sum_{p=0}^{H-1} \sum_{q=0}^{W-1} v(p, q) \cdot \omega_H^{-p \cdot h} \omega_W^{-q \cdot w},
$$
$$
\forall (h, w) \in \{0, \ldots, H-1\} \times \{0, \ldots, W-1\}. \quad (83)
$$

Analogous to Fact 9, $\boldsymbol{F}$ relates $\text{DFT}(\boldsymbol{z})$ and $\text{circ}(\boldsymbol{z})$ as follows.

**Fact 9 (2D-DFT are eigenvalues of the doubly block circulant matrix)** *Given a signal $\boldsymbol{z} \in \mathbb{C}^{H \times W}$, we have*

$$
\boldsymbol{F} \cdot \text{circ}(\boldsymbol{z}) \cdot \boldsymbol{F}^* = \text{diag}(\text{vec}(\text{DFT}(\boldsymbol{z}))) \quad \text{or} \quad \text{circ}(\boldsymbol{z}) = \boldsymbol{F}^* \cdot \text{diag}(\text{vec}(\text{DFT}(\boldsymbol{z}))) \cdot \boldsymbol{F}. \quad (84)
$$

**Doubly block circulant matrix and 2D-DFT for multi-channel signals.** We now consider multi-channel 2D signals $\bar{\boldsymbol{z}} \in \mathbb{R}^{C \times H \times W}$. Let $\text{DFT}(\bar{\boldsymbol{z}}) \in \mathbb{C}^{C \times H \times W}$ be a matrix where the $c$-th slice on the first dimension is the DFT of the corresponding signal $\boldsymbol{z}[c]$. That is, $\text{DFT}(\bar{\boldsymbol{z}})[c] = \text{DFT}(\boldsymbol{z}[c]) \in \mathbb{C}^{H \times W}$. We use $\text{DFT}(\bar{\boldsymbol{z}})(p, q) \in \mathbb{C}^C$ to denote slicing of $\bar{\boldsymbol{z}}$ on the frequency dimensions.

By using Fact 9, $\text{circ}(\bar{z})$ and $\text{DFT}(\bar{z})$ are related as follows:

$$\text{circ}(\bar{z}) = \begin{bmatrix} \boldsymbol{F}^* \cdot \text{diag}(\text{vec}(\text{DFT}(\boldsymbol{z}[1]))) \cdot \boldsymbol{F} \\ \vdots \\ \boldsymbol{F}^* \cdot \text{diag}(\text{vec}(\text{DFT}(\boldsymbol{z}[C]))) \cdot \boldsymbol{F} \end{bmatrix}$$

$$= \begin{bmatrix} \boldsymbol{F}^* & \cdots & \boldsymbol{0} \\ \boldsymbol{0} & \cdots & \boldsymbol{0} \\ \vdots & \ddots & \vdots \\ \boldsymbol{0} & \cdots & \boldsymbol{F}^* \end{bmatrix} \cdot \begin{bmatrix} \text{diag}(\text{vec}(\text{DFT}(\boldsymbol{z}[1]))) \\ \text{diag}(\text{vec}(\text{DFT}(\boldsymbol{z}[2]))) \\ \vdots \\ \text{diag}(\text{vec}(\text{DFT}(\boldsymbol{z}[C]))) \end{bmatrix} \cdot \boldsymbol{F}. \quad (85)$$

Similar to the 1D case, this relation can be leveraged to produce a fast implementation of ReduNet in the spectral domain.

**Translation-invariant ReduNet in the Spectral Domain.** Given a collection of multi-channel 2D signals $\bar{\boldsymbol{Z}} \in \mathbb{R}^{C \times H \times W \times m}$, we denote

$$\text{DFT}(\bar{\boldsymbol{Z}})(p,q) \doteq [\text{DFT}(\bar{z}^1)(p,q), \ldots, \text{DFT}(\bar{z}^m)(p,q)] \quad \in \mathbb{R}^{C \times m}. \quad (86)$$

We introduce the notations $\bar{\mathcal{E}}(p,q) \in \mathbb{R}^{C \times C \times H \times W}$ and $\bar{\mathcal{C}}^j(p,q) \in \mathbb{R}^{C \times C \times H \times W}$ given by

$$\bar{\mathcal{E}}(p,q) \doteq \alpha \cdot \left[ \boldsymbol{I} + \alpha \cdot \text{DFT}(\bar{\boldsymbol{Z}})(p,q) \cdot \text{DFT}(\bar{\boldsymbol{Z}})(p,q)^* \right]^{-1} \in \mathbb{C}^{C \times C}, \quad (87)$$

$$\bar{\mathcal{C}}^j(p,q) \doteq \alpha_j \cdot \left[ \boldsymbol{I} + \alpha_j \cdot \text{DFT}(\bar{\boldsymbol{Z}})(p,q) \cdot \boldsymbol{\Pi}_j \cdot \text{DFT}(\bar{\boldsymbol{Z}})(p,q)^* \right]^{-1} \in \mathbb{C}^{C \times C}. \quad (88)$$

In above, $\bar{\mathcal{E}}(p,q)$ (resp., $\bar{\mathcal{C}}^j(p,q)$) is the $(p,q)$-th slice of $\bar{\mathcal{E}}$ (resp., $\bar{\mathcal{C}}^j$) on the last two dimensions. Then, the gradient of $\Delta R_{\text{circ}}(\bar{\boldsymbol{Z}}, \boldsymbol{\Pi})$ with respect to $\bar{\boldsymbol{Z}}$ can be calculated by the following result.

**Theorem C.1 (Computing multi-channel 2D convolutions $\bar{E}$ and $\bar{C}^j$)** *Let* $\bar{\boldsymbol{U}} \in \mathbb{C}^{C \times H \times W \times m}$ *and* $\bar{\boldsymbol{W}}^j \in \mathbb{C}^{C \times H \times W \times m}, j = 1, \ldots, k$ *be given by*

$$\bar{\boldsymbol{U}}(p,q) \doteq \bar{\mathcal{E}}(p,q) \cdot \text{DFT}(\bar{\boldsymbol{Z}})(p,q), \quad (89)$$

$$\bar{\boldsymbol{W}}^j(p,q) \doteq \bar{\mathcal{C}}^j(p,q) \cdot \text{DFT}(\bar{\boldsymbol{Z}})(p,q), \quad j = 1, \ldots, k, \quad (90)$$

*for each* $(p,q) \in \{0, \ldots, H-1\} \times \{0, \ldots, W-1\}$. *Then, we have*

$$\frac{1}{2HW} \frac{\partial \log \det(\boldsymbol{I} + \alpha \cdot \text{circ}(\bar{\boldsymbol{Z}})\text{circ}(\bar{\boldsymbol{Z}})^*)}{\partial \bar{\boldsymbol{Z}}} = \text{IDFT}(\bar{\boldsymbol{U}}), \quad (91)$$

$$\frac{1}{2HW} \frac{\partial \left( \gamma_j \log \det(\boldsymbol{I} + \alpha_j \cdot \text{circ}(\bar{\boldsymbol{Z}})\bar{\boldsymbol{\Pi}}^j \text{circ}(\bar{\boldsymbol{Z}})^*) \right)}{\partial \bar{\boldsymbol{Z}}} = \gamma_j \cdot \text{IDFT}(\bar{\boldsymbol{W}}^j \boldsymbol{\Pi}^j). \quad (92)$$

This result shows that the calculation of the derivatives for the 2D case is analogous to that of the 1D case. Therefore, the construction of the ReduNet for 2D translation invariance can be performed using Algorithm 1 with straightforward extensions.

# D IMPLEMENTATION DETAILS AND ADDITIONAL EXPERIMENTS

Code for reproducing the results in this work will be made publicly available with the publication of this paper. Disclaimer: in this work we do not particularly optimize any of the hyper parameters, such as the number of initial channels, kernel sizes, and learning rate etc., for the best performance. The choices are mostly for convenience and just minimally adequate to verify the concept, due to limited computational resource.

## D.1 ADDITIONAL EXPERIMENTS ON LEARNING MIXTURE OF GAUSSIANS IN $\mathbb{S}^1$ AND $\mathbb{S}^2$

We provide the cosine similarity results for the experiments described in Figure 2. The results are shown in Figure 4. We can observe that the network can map the data points to orthogonal subspaces.

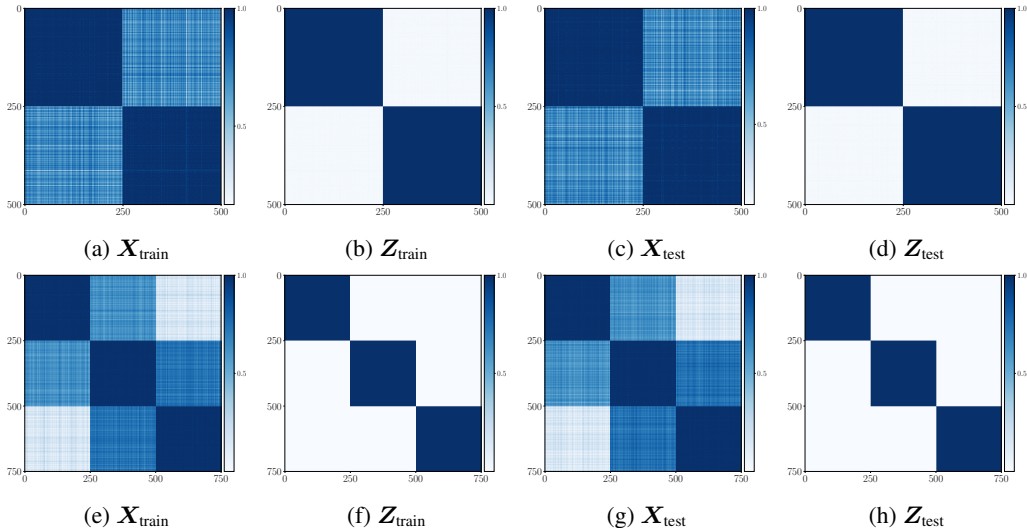

(a) $\boldsymbol{X}_{\text{train}}$ (b) $\boldsymbol{Z}_{\text{train}}$ (c) $\boldsymbol{X}_{\text{test}}$ (d) $\boldsymbol{Z}_{\text{test}}$

(e) $\boldsymbol{X}_{\text{train}}$ (f) $\boldsymbol{Z}_{\text{train}}$ (g) $\boldsymbol{X}_{\text{test}}$ (h) $\boldsymbol{Z}_{\text{test}}$

Figure 4: Cosine similarity (absolute value) for $2D$ and $3D$ Mixture of Gaussians. Lighter color implies samples are more orthogonal.

**Additional experiments on $\mathbb{S}^1$ and $\mathbb{S}^2$.** We also provide additional experiments on learning mixture of Gaussians in $\mathbb{S}^1$ and $\mathbb{S}^2$ in Figure 5. We can observe similar behavior of the proposed ReduNet: the network can map data points from different classes to orthogonal subspaces.

**Additional experiments on $\mathbb{S}^1$ with more than 2 classes.** We try to apply ReduNet to learn mixture of Gaussian distributions on $\mathbb{S}^1$ with the number of class is larger than 2. Notice that these are the cases to which the existing theory about MCR$^2$ (Yu et al., 2020) no longer applies. These experiments suggest that the MCR$^2$ still promotes between-class discriminativeness with so constructed ReduNet. In particular, the case on the left of Figure 6 indicates that the ReduNet has "merged" two linearly correlated clusters into one on the same line. This is consistent with the objective of rate reduction to group data as linear subspaces.

## D.2 EXPERIMENTS ON UCI DATASETS

We evaluate the proposed ReduNet on some real datasets, namely the two UCI tasks (Dua & Graff, 2017): iris and mice. There are 3 classes in iris dataset and the number of features is 4. For mice dataset, there are 8 classes and the number of features is 82. We randomly select 70% data as the training data, and use the rest for evaluation. The results are summarized in Table 1. We compare our method with logistic regression, SVM, and random forest, and we use the implementations by **sklearn** (Pedregosa et al., 2011). From Table 1, we find that the forward-constructed ReduNet is able to achieve comparable performance with classic methods such as logistic regression, SVM, and random forest.

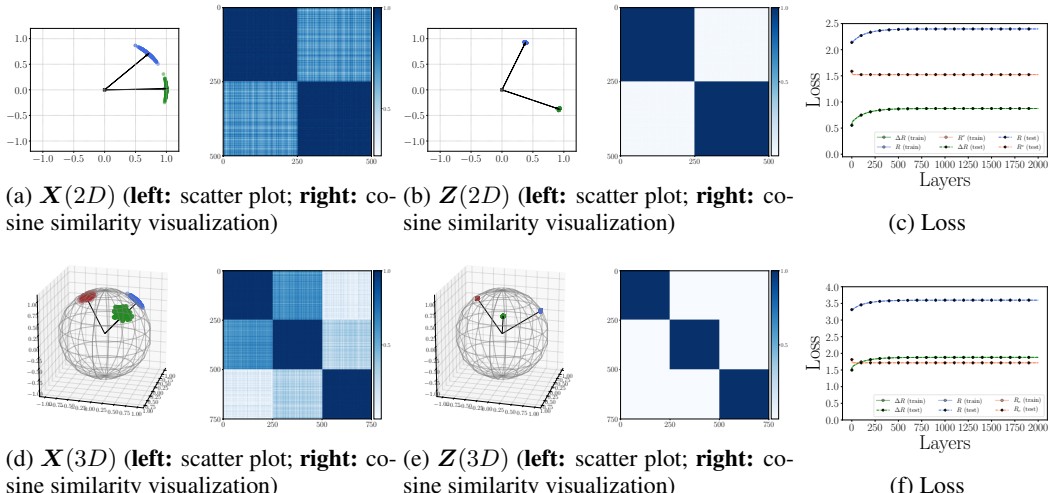

(a) $\boldsymbol{X}(2D)$ (**left:** scatter plot; **right:** cosine similarity visualization)

(b) $\boldsymbol{Z}(2D)$ (**left:** scatter plot; **right:** cosine similarity visualization)

(c) Loss

(d) $\boldsymbol{X}(3D)$ (**left:** scatter plot; **right:** cosine similarity visualization)

(e) $\boldsymbol{Z}(3D)$ (**left:** scatter plot; **right:** cosine similarity visualization)

(f) Loss

Figure 5: Learning mixture of Gaussians in $\mathbb{S}^1$ and $\mathbb{S}^2$. (**Top**) For $\mathbb{S}^1$, we set $\sigma_1 = \sigma_2 = 0.1$; (**Bottom**) For $\mathbb{S}^2$, we set $\sigma_1 = \sigma_2 = \sigma_3 = 0.1$.

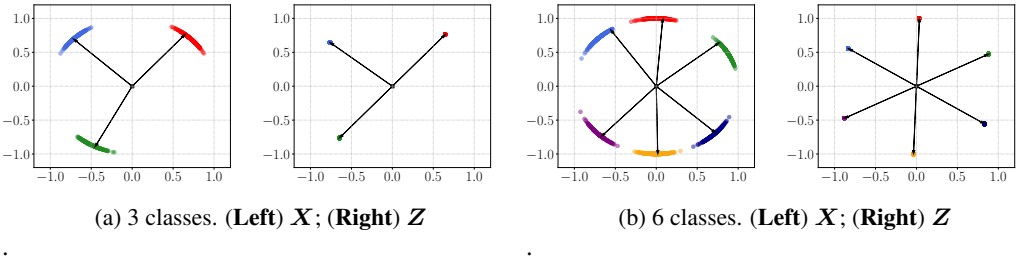

(a) 3 classes. (**Left**) $\boldsymbol{X}$; (**Right**) $\boldsymbol{Z}$

(b) 6 classes. (**Left**) $\boldsymbol{X}$; (**Right**) $\boldsymbol{Z}$

Figure 6: Learning mixture of Gaussian distributions with more than 2 classes. For both cases, we use step size $\eta = 0.5$ and precision $\epsilon = 0.1$. For (a), we set iteration $L = 2,500$; for (b), we set iteration $L = 4,000$.

### D.3 ADDITIONAL EXPERIMENTS ON LEARNING SHIFT INVARIANT FEATURES

We provide additional experiments for *Learning Shift Invariant Features* in §3. The code for sampling from $h_1(t) = \mathsf{sin}(t) + \epsilon$ and $h_2(t) = \mathsf{sign}(\mathsf{sin}(t)) + \epsilon$ is described in Algorithm 2, and the pseudocode for sampling from 2 classes $\{h_1, h_2\}$ is described as follows, we sample training and test signals using the same procedure.

```
t0 = np.random.uniform(low=0, high=10*np.pi, size=samples)
x = np.linspace(t0, t0+2*np.pi, time).T
noise1 = np.random.normal(0, 0.1, size=(samples, time))
X1 = np.sin(x) + noise1
noise2 = np.random.normal(0, 0.1, size=(samples, time))
X2 = np.sign(np.sin(x)) + noise2
data = np.vstack([X1, X2])
labels = np.hstack([np.ones(samples)*1,
                    np.ones(samples)*2]).astype(np.int32)
```

We also provide cosine similarities between samples in Figure 8. We visualize the cosine similarities for the input $\boldsymbol{X}_{\text{train}}, \boldsymbol{X}_{\text{test}}$ as well as the learned representations $\boldsymbol{Z}_{\text{train}}, \boldsymbol{Z}_{\text{test}}$. The cosine similarity between sample pairs selected from different classes are shown in Figure 9. We can observe that the original data is not orthogonal w.r.t. different classes, and the the ReduNet is able to learn discriminative (orthogonal) representations.

Table 1: Performance (Accuracy) on iris and mice of the UCI datasets.

|  | REDUNET | LOGISTIC REGRESSION | SVM | RANDOM FOREST |
|---|---|---|---|---|
| IRIS | 0.978 | 0.933 | 0.933 | 0.978 |
| MICE | 0.972 | 0.855 | 0.975 | 0.985 |

---

**Algorithm 2** Pseudocode for sampling signals from $1D$ functions

---

**Input:** Number of samples $m$, number of classes $k$, number of features $n$, function $\{h_1, \ldots, h_k\}$.
1: **for** $j = 1, 2, \ldots, k$ **do**
2:     **for** $i = 1, 2, \ldots, m$ **do**
3:        $t_0 \sim \text{Uniform}[0, 10\pi]$;
4:        $\boldsymbol{t} = [t_0, t_0 + 2\pi/n, t_0 + (2\pi/n) \cdot 2, t_0 + (2\pi/n) \cdot 3, \ldots, t_0 + (2\pi/n) \cdot (n-1)]$;
5:        $\boldsymbol{x}_j^i = h_j(\boldsymbol{t}) + \epsilon$ *# broadcast over vector $\boldsymbol{t}$*;
6:     **end for**
7:     $\boldsymbol{X}_j = [\boldsymbol{x}_j^1, \boldsymbol{x}_j^2, \ldots, \boldsymbol{x}_j^m]$;
8: **end for**
9: $\boldsymbol{X} = [\boldsymbol{X}_1, \boldsymbol{X}_2, \ldots, \boldsymbol{X}_k]$;
10: shuffle $\boldsymbol{X}$.
**Output:** outputs $\boldsymbol{X}$.

---

## D.4   ADDITIONAL EXPERIMENTS ON LEARNING ROTATIONAL INVARIANCE ON MNIST

We provide additional experiments for *learning rotational invariance on MNIST* in §3. Examples of rotated images are shown in Figure 12. We compare the accuracy (both on the original test data and the shifted test data) of the ReduNet (without considering invariance) and the shift invariant ReduNet. For ReduNet (without considering invariance), we use the same training dataset as the shift invariant ReduNet, we set iteration $L = 3, 500$, step size $\eta = 0.5$, and precision $\epsilon = 0.1$. The results are summarized in Table 2. With the invariant design, we can see from Table 2 that the shift invariant ReduNet achieves better performance in terms of invariance on the MNIST binary classification task.

We also provide cosine similarities between samples in Figure 10. We visualize the cosine similarities for the input $\boldsymbol{X}_{\text{train}}$, $\boldsymbol{X}_{\text{test}}$ as well as the learned representations $\boldsymbol{Z}_{\text{train}}$, $\boldsymbol{Z}_{\text{test}}$. The cosine similarity between sample pairs selected from different classes are shown in Figure 11. We can observe that the constructed ReduNet is able to learn discriminative (orthogonal) and invariant representations for MNIST digits.

Table 2: Comparing network performance on learning rotational-invariant representations on MNIST.

|  | REDUNET | REDUNET (SHIFT-INVARIANT) |
|---|---|---|
| ACC (ORIGINAL TEST DATA) | 0.983 | 0.996 |
| ACC (TEST DATA WITH ALL POSSIBLE SHIFTS) | 0.707 | 0.993 |

## D.5   EXPERIMENTS ON LEARNING 2D TRANSLATION INVARIANCE ON MNIST

In this part, we provide experimental results for verifying the invariance property of ReduNet under 2D translations. We construct 1). ReduNet (without considering invariance) and 2). 2D translation-invariant ReduNet for classifying digit '0' and digit '1' on MNIST dataset. We use $m = 1,000$ samples (500 samples from each class) for training the models, and use another 500 samples (250 samples from each class) for evaluation. To evaluate the 2D translational invariance, for each test image $\boldsymbol{x}_{\text{test}} \in \mathbb{R}^{H \times W}$, we consider *all* translation augmentations of the test image with a stride=7. More specifically, for the MNIST dataset, we have $H = W = 28$. So for each image, the total number of all cyclic translation augmentations (with stride=7) is $4 \times 4 = 16$. Examples of translated images are shown in Figure 13. Notice that such translations are considerably larger than normally considered in the literature since we consider invariance to the entire group of cyclic translations on the $H \times W$ grid as a torus. See Figure 13 for some representative test samples.

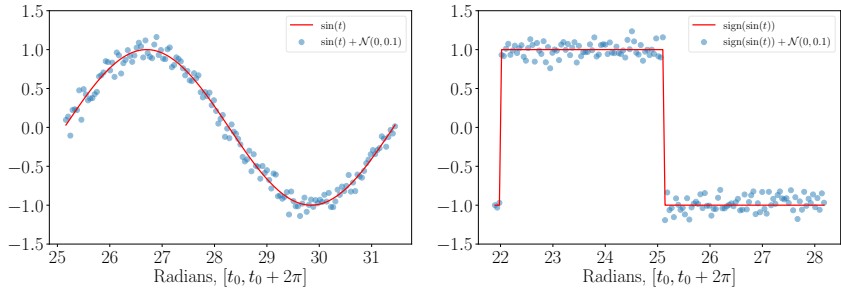

Figure 7: Visualization of signals in 1D. Blue dots represent the sampled signal used for training with dimension $n = 150$. Red curves represent the underlying 1D function (noiseless). **(Left)** One sample from class 1; **(Right)** One sample from class 2.

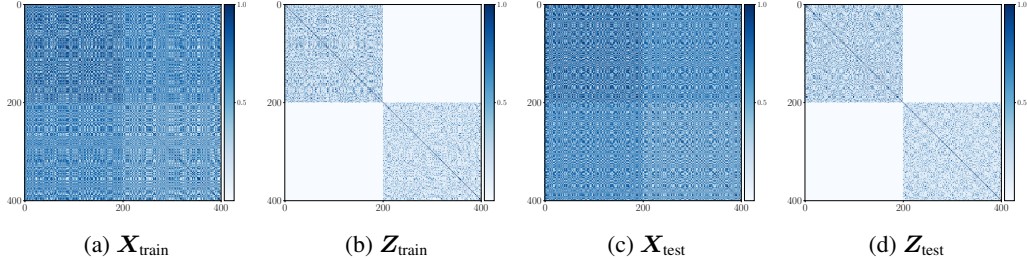

(a) $\boldsymbol{X}_{\text{train}}$      (b) $\boldsymbol{Z}_{\text{train}}$      (c) $\boldsymbol{X}_{\text{test}}$      (d) $\boldsymbol{Z}_{\text{test}}$

Figure 8: Cosine similarity (absolute value) of training/test data as well as training/test representations for learning 1D functions.

For ReduNet (without considering translation invariance), we set iteration $L = 2,000$, step size $\eta = 0.1$, and precision $\epsilon = 0.1$. For translation-invariant ReduNet, we set $L = 2,000$, step size $\eta = 0.5$, precision $\epsilon = 0.1$, number of channels $C = 5$, and kernel size is set as $3 \times 3$. We summarize the results in Table 3. Similar to the 1D rotational results on the MNIST dataset, the translation-invariant ReduNet achieves better performance under translations compared with the RedeNet without considering invariance. The accuracy drop of the translation-invariant ReduNet is much less than the one of ReduNet without invariance design.

Table 3: Comparing network performance on learning 2D translation-invariant representations on MNIST.

|  | REDUNET | REDUNET (TRANSLATION-INVARIANT) |
|---|---|---|
| ACC (ORIGINAL TEST DATA) | 0.980 | 0.975 |
| ACC (TEST DATA WITH ALL POSSIBLE SHIFTS) | 0.540 | 0.909 |

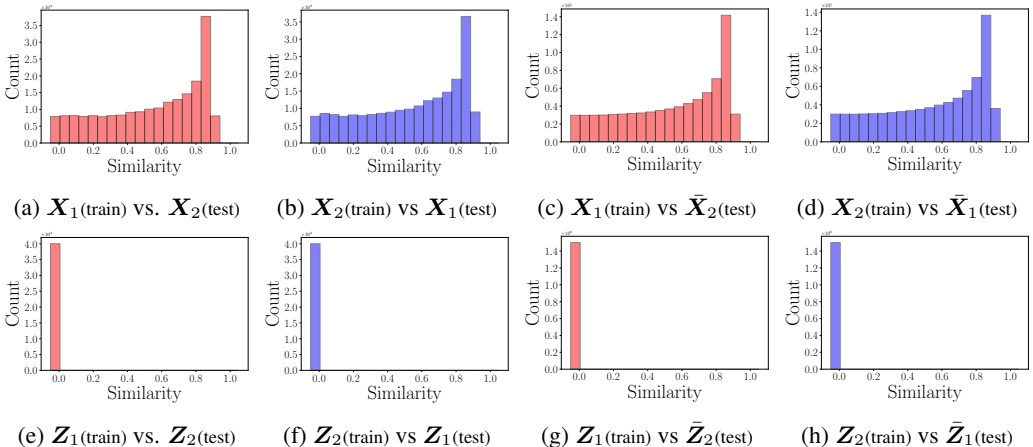

Figure 9: Histogram of cosine similarity between pairs sampled from different classes for learning 1D function. The histogram of cosine similarity between training data $X_c$ as well as representations $Z_c$ vs. testing (shifted) data $\bar{X}_{c'}$ as well as (shifted) representations $\bar{Z}_{c'}$, where we let $c$ denote the class index and $c \neq c'$.

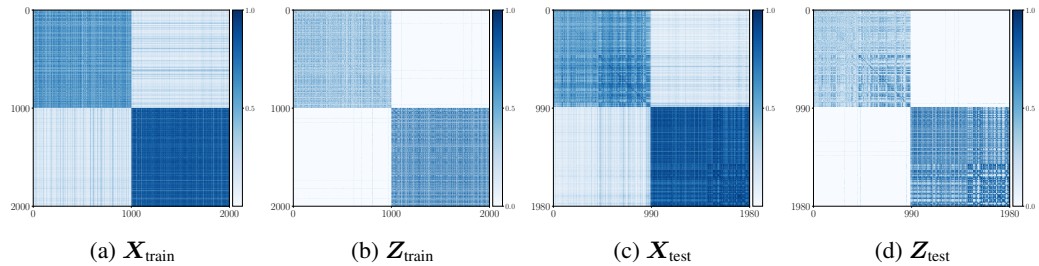

Figure 10: Cosine similarity (absolute value) of training/test data as well as traning/test representations for learning rotational invariant representations on MNIST.

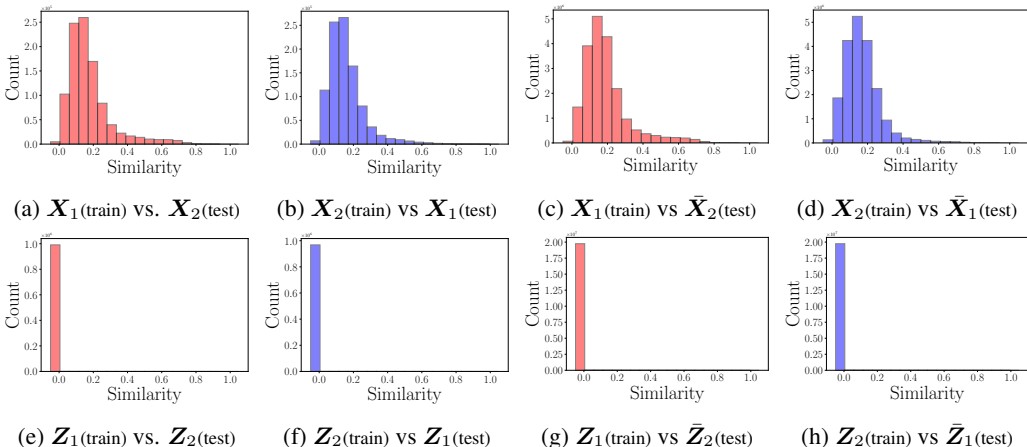

Figure 11: Histogram of cosine similarity between pairs sampled from different classes for learning rotational invariant representations on MNIST. The histogram of cosine similarity between training data $X_c$ as well as representations $Z_c$ vs. testing (shifted) data $\bar{X}_{c'}$ as well as (shifted) representations $\bar{Z}_{c'}$, where we let $c$ denote the class index and $c \neq c'$.

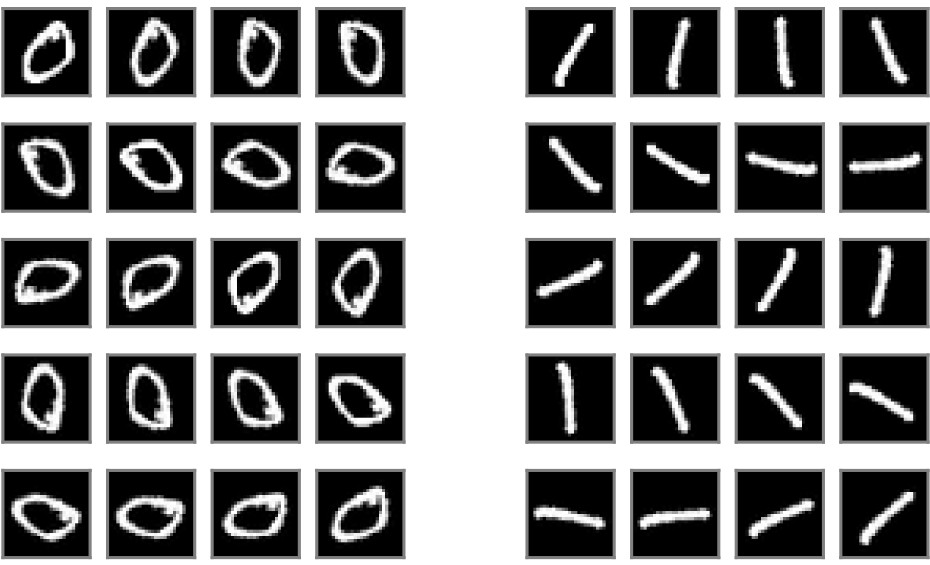

Figure 12: Examples of rotated images of MNIST digits for testing rotation invariance, each rotated by 18°. (**Left**) digit '0'. (**Right**) digit '1'.

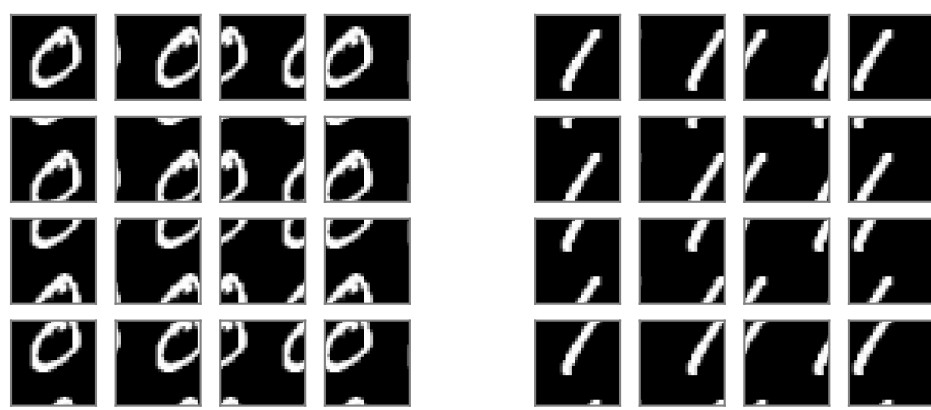

Figure 13: Examples of translated images of MNIST digits (with stride=7) for testing cyclic translation invariance of the ReduNet. (**Left**) digit '0'. (**Right**) digit '1'.

