# OpenReview forum: "Deep Networks from the Principle of Rate Reduction"
_ICLR.cc/2021/Conference — Reject_

### Official Review · AnonReviewer4 · 2020-10-27
**MCR^2 principle makes sense, details of ReduNet are confusing**

**Rating:** 6
**Confidence:** 3

**Review:**

The paper formulates an iterative process of deriving encoding of data into feature space as a deep model, called ReduNet, where each layer corresponds to iteration of the optimisation process the feature space according to the MCR^2 principle.  The MCR^2 optimisation maps points of different classes into separate subspaces, with volume of each subspace being minimized while the volume of the entire space is maximized.   It is analogous to pushing like things together and unlike things apart.  The novelty of the paper is in that formulation of the feature optimisation is baked-in into a deep architecture.

MCR^2 principle seems like a sensible approach to learning, especially given that embedding algorithms (such as face encoding) use it already.  It’s nice to see some rigorous mathematical treatment on this.  However, I get confused pretty early on by the notations.  If f(x,\theta)\in \mathcal{R}^k and z=f(x)\in \mathcal{R}^n….then since y=h(z), then f(x,\theta)=h(f(x))…and so f(x,\delta) and f(x) are two different functions.  Yet later in the text z=f(x,\theta).  And from then on, including equation 11, f(x,\delta)=\psi^L(z_L+\etag(z_{L-1},\theta_{L-1})…which would make it seem f(x,\theta)\in \mathca{R}^n.  And what is g(z_l,\theta_1)?  Equation 8 tells us what g(z_l,\theta_1) must approximate…but what is it exactly?   Is that a neural network, or some model, with parameters \theta_l?  Or is Equation 8 a definition of g(z_l,\theta_l)…in which case what is \theta_l?  I don’t think the math is necessarily wrong…just notation is confusing and definitions changing/not consistent.

I have also questions about equation 11, where number of layers is equivalent to iterations while maximizing MCR^2 and the width of each layer corresponds to m, the training points in the dataset.  So, in order to do a mapping of an input x, we need to perform L iterative steps using the entire m points every time?  Isn’t that equivalent to doing a massive learning process, using the entire dataset, for each mapping?  How computationally costly is that?  I also don’t quite understand how \psi^l(z_1,\theata_1) works - how does \theta_l change over iterations?  Experimental section is not helping me with this, since it’s stated that E, C^j are computed for each layer…but there are no details on what \theta_l is and how g(z_1,\theta_1) is evaluated.  And if f(x,\theta)=z^L…then how do we get classification from that?  Is it just based on definition of \hat{\pi}^j(z_l) from page 4?

Finally, I am not sure if the result of obtaining a convnet architecture in ReduNet when translation invariance constraint is added the embedding is all that surprising.  Isn't it somewhat obvious that if each layer of ReduNet is invariant in some way, then the entire network is invariant?  It feels like that what we are learning here is not that in order to have translation-invariant mapping we must have a convent...but rather that we can obtain a translation invariant deep architecture with translation invariant layers.

---

> ### Author Response · Authors · 2020-11-13
> **Response to AnonReviewer4**
>
> Thank you for your feedback on our work.
>
> **Q: Question regarding $f(x, \theta) \in \mathbb{R}^k$ vs. $f(x, \theta) \in \mathbb{R}^n$.**
>
> **A:** The notation $f(x, \theta) \in \mathbb{R}^k$ is used ONLY in the third line of Sec. 2 to denote a typical end-to-end deep neural network that maps data to label. Throughout the rest of the paper where we describe the proposed ReduNet, we always use the notation $f(x, \theta) \in \mathbb{R}^n$ to denote a mapping from data space to feature space. This is a slight abuse of notation. We apologize for the confusion, and will make adjustment in our writing to make it clear.
>
> **Q: Question regarding the definition of $g(z_\ell, \theta_\ell)$.** ''And what is $g(z_l,\theta_1)$?  Equation 8 tells us what $g(z_l,\theta_1)$ must approximate...but what is it exactly? Is that a neural network, or some model, with parameters $\theta_l$?  Or is Equation 8 a definition of $g(z_l,\theta_l)$... in which case what is $\theta_l$?''
>
> **A:** In Eq. (8), we introduced a definition of $g(z_\ell, \theta_\ell)$ from emulating gradient ascent on rate reduction. We then argued that Eq. (8) cannot be evaluated on test data, hence we used Eq. (9) to replace the second term of Eq. (8).
> Therefore, $g(z_\ell, \theta_\ell)$ is ultimately defined as Eq. (8) with the second term replaced by Eq. (9). Correspondingly, the parameter $\theta_\ell$ contains the matrices $E_\ell$ and $C_{\ell}^j$ for $j = 1, \cdots, k$, as stated in the paragraph beneath Eq. (10).
>
>
> **Q: I have also questions about equation 11, where number of layers is equivalent to iterations while maximizing MCR$^2$ and the width of each layer corresponds to m, the training points in the dataset.**
>
> **A:** The width of each layer is proportional to the number of classes $k$ as well as the dimension of the input space $n$, which is not related to the number of training points $m$.
>
> **Q: Question regarding Eq. (11).**  ''So, in order to do a mapping of an input $x$, we need to perform $L$ iterative steps using the entire $m$ points every time?''
>
> **A:** Eq. (11) represents the ReduNet learned from $m$ training data, which is a mapping from $x$ to $f(x, \theta)$, where $\theta$ is the network parameter. Performing such a mapping for test data does not require the entire $m$ points, since $\theta$ has already been learned in the training phase and is fixed.
>
> **Q: Regarding computational complexity.**
>
> **A:** To construction the training layer by layer, we use all the training samples in this paper to compute matries $\{E_{\ell}, C_{\ell}^{j}\}$ of the $\ell$-th layer. During the test, we only need to forward through the network, no additional computation is needed during test. Also, we only compute the matries $E_{\ell}$, $C_{\ell}^{j}$ (for $j \in [k]$, $\ell \in [L]$) once and no back propergation is needed here. Also, ReduNets are trained only with forward initialization and without back-propagation.
>
> **Q:** ''I also don’t quite understand how $\psi^l(z_1, \theta_1)$ works - how does $\theta_l$ change over iterations? Experimental section is not helping me with this, since it’s stated that $E, C^j$ are computed for each layer...but there are no details on what $\theta_l$ is and how $g(z_1,\theta_1)$ is evaluated.''
>
> **A:** First of all, in Eq. (10), we use $\theta_{\ell}$ to denote the parameters in the $\ell$-th layer ($E_{\ell}$ and $C_{\ell}^{j}$ where $j \in [k]$) Once we computed/initialize $E_{\ell}, C_{\ell}^{j}$ by Eq. (4) and Eq. (5), $\theta_{\ell}$ will not change.   $g(z_{\ell},\theta_{\ell})$ is defined in Eq. (8) where the second term is approximated by Eq. (9).
>
> **Q:** ''And if $f(x,\theta)=z^L$... then how do we get classification from that? Is it just based on definition of $\hat{\pi}^j(z_l)$ from page 4?''
>
> **A:** For the output of the $L$-th layer $f(x,\theta)=z^L$, we apply nearest subspace classifier for classification, which is described in the paragraph after Eq. (11).
>
>
> **Q: Convnet architecture vs. translation invariance.**
>
> **A:** Unlike previous neural networks in which convolution is designed for each layer to obtain invariance representations, the proposed ReduNet is designed from the principle of obtaining an invariance feature mapping from which convolution arises from the learning objective of rate reduction. While previous works have constructed invariance networks by stacking invariance layers (i.e., convolutional layers), there is still a lack of understanding on how a network should be constructed: questions pertaining the role of multiple layers, role of nonlinear activation, relationship among multiple convolutional kernels and channels, etc., remain unanswered. In contrast, this work proposes to construct a neural network from the principle of maximizing rate reduction. Not only is the entire neural network (rather than only the convolutional layers) is constructed in a principled way, but also the learned features have the desired invariance and discriminative properties.

---

> ### Author Response · Authors · 2020-11-16
> **Response to AnonReviewer4 [Additional experiments regarding computational complexity]**
>
> **Q: Regarding computational complexity.** ''So, in order to do a mapping of an input x, we need to perform L iterative steps using the entire m points every time? Isn’t that equivalent to doing a massive learning process, using the entire dataset, for each mapping? How computationally costly is that?''
>
> **A:** We did some preliminary experiments on speeding up the initialization of ReduNet. From our previous experiments, we used the entire training set to initialize the weight matrices, $\{E_{\ell}, C_{\ell}^{1}, \dots, C_{\ell}^{k}\}$, for $\ell \in [L]$. Alternatively, we could apply stochastic mini-batch training samples to speed up the initialization of ReduNet. We present our results on the MNIST dataset using ReduNet, where our training set contains 1,000 (vectorized) samples for digit '0' and digit '1' each. Then the weights of every layer are initialized using mini-batch samples randomly sampled from the 2,000 training samples. As shown in the following Table, by applying this mini-batch initialization scheme, we are able to achieve similar performances as using all training samples to initialize the network. One benefit of this initialization scheme is that it can improve the computational complexity for computing the network weights.
>
>
> | Batch Size              	             	| Accuracy  	|
> |-----------------------------------------|-----------------|
> |100   		   		   	       |  98.0%   	|
> |200    		   		   	       |  98.4%   	|
> |500   		   		   	       |  98.2%   	|
> |1,000   		   		   	       |  97.5%   	|
> |2,000 (*all samples*) &nbsp; |  97.2%   	|

---

### Official Review · AnonReviewer3 · 2020-10-28
**The paper is well-founded and presented with an innovative deep network perspective**

**Rating:** 9
**Confidence:** 3

**Review:**

The authors propose a deep network approach using the principle of rate reduction as a loss under a gradient ascent approach, avoiding traditional backpropagation. Besides, the work attempts to interpret the proposed framework from both geometrical and statistical views. Then, shift-invariant properties are discussed. The innovative method allows the inclusion of a new layer structure named ReduNet, which could benefit the deep learning community. Though the experiments are not challenging concerning the studied databases, the authors aim to probe the concept without a complete implementation tuning. Overall, the paper is illustrative enough regarding the mathematical foundation.

---

> ### Author Response · Authors · 2020-11-13
> **Response to AnonReviewer3**
>
> Thank you very much for appreciating the novelty and significance of our paper. As the reviewer has correctly pointed out, our work aims to ''prove the concept without a complete implementation tuning''. The current construction is very basic and idealistic (by choice). There are many aspects of the ReduNet that can be improved or made significantly more scalable and practical. We will explore these directions in our future work.

---

### Official Review · AnonReviewer2 · 2020-10-29
**interesting paper with solid analysis**

**Rating:** 6
**Confidence:** 2

**Review:**

The paper proposed a network derived from maximizing of rate reduction, known as the MCR^2 principle, where  all parameters are explicitly constructed layer by layer in a forward propagation fashion. Compared with exiting work by Yu et al. (2020), the proposed work is more like a "white box" that each layer is more interpretable.  Results showed the proposed network can learn a good discriminative deep representation without any back propagation training. The derivation of the network also suggests that the network is more efficient to learn and construct in the spectral domain.

Overall, the proposed work looks reasonable. The paper is well-structured. The derivation and experiments seem convincing.  Unfortunately, the proposed work is out of the reviewer's expertise and therefore it is hard to provide valuable comments.

---

> ### Author Response · Authors · 2020-11-13
> **Response to AnonReviewer2**
>
> We thank the reviewer for the positive comments on our method, presentation, and experimental design of our submission. Indeed, the proposed approach represents a significant departure from existing approaches to studying deep networks. As the work intends to interpret deep networks as a white box, it does call upon results from many other but related domains such as data compression, signal processing, and optimization.

---

### Official Review · AnonReviewer1 · 2020-10-30
**Novel perspective for deriving network architecture.**

**Rating:** 6
**Confidence:** 4

**Review:**

#### Summary
This paper proposes a theoretical understanding of neural architecture using the  principle of rate reduction. Yu et.al 2020 and the derived optimization steps naturally leads to operations such as network layers and identity residual. By enforcing shift invariant, it can also lead to convolutional operation. The network can be constructed with a forward propagation fasion, which conserves good discriminative ability.

#### Novelty
Theorectical guidance of network design is one of the key direction in representation learning.
The paper proposes a novel perspective (rate reduction) in network construction that generalized to the design of networks such as resnet and resnext. However, there are a bunch or other networks such as denseNet, non-local network etc., which are also performing well in practice and designed with huerestics of larger context and better gradient propagation, will the author also include these representations within the objective of rate reduction?

Or does the generation process from rate reduction (compact discriminative representation) come with additional guidance of what network, or answer the question in introduction, what is the object is optimal for network design, which could show to be effective on multiple tasks.

I think the major issue of current result is that we can see the objective explain  that the set up of networks (resNet) seeks a compact representation, but the author has not shown strong experiments that their yielded alternatives by optimizing the objective (rate reduction) has positive relation with the network generalization. Is higher rate reduction leads network with better performance on multiple tasks ?

Last, it is obvious that shift invariance leads to convolution, which is effective for classification of 2D objects. However, in realistic senario, we may seem equivalent disentangled representation rather  than invariance, is it possible for the objective leads to convolution without explicit inducing shift invariance ?

#### Writing and reference

The writing is good and easy to follow, checked few derivatives which are correct. I think the paper could also related to optimization inspired network design, e.g. Optimization Algorithm Inspired Deep Neural Network Structure Design, which delivers another perspective.


In general, I think the objective is novel, but not generalized enough to explain lots of high performance networks yet. However more exploration of theoretical study should be encouraged.

---

> ### Author Response · Authors · 2020-11-13
> **Response to AnonReviewer1**
>
> Thank your for your positive comments and feedback on our work.
>
> **Q: Explaining DenseNet.**
>
> **A:** Our framework does provide a possible interpretation for DenseNet as is discussed in Remark 4 (see Appendix A).
> In particular, since our network is constructed by emulating the basic gradient scheme, the additional skip connections as in DenseNet can be interpreted as emulating accelerated gradient methods such as the Nesterov acceleration.
> Hence, one may attribute the superior performance of DenseNet as accelerating the convergence of gradient descent so that the number of layers can be significantly reduced.
> While our framework provides possible explanations to many popular network architectures and building blocks, it is far from explaining all networks that have been found effective in the vast literature. But this work points out a new direction for a similar explanation for other networks as well.
>
>
> **Q: Optimal architectural design.** ''what is the object is optimal for network design''
>
> **A:** This work proposes to design neural networks from the principle of maximizing rate reduction. By this principle, the optimal network should be such that the learned representations achieve the desired discriminative and invariance properties as measured by the rate reduction objective. Hence, the depth is determined by the convergence of rate reduction, as we did in our experiments; meanwhile, the width (i.e., the number of filter banks) of the network is determined by the task in a principled way, from the number of classes to be learned from the data.
>
>
> **Q: Relation between rate reduction and generalization.** ''Is higher rate reduction leads network with better performance on multiple tasks?''
>
> **A:** The benefits of optimizing rate reduction for obtaining better generalization performance are not the main point nor claimed contributions of this paper, but it has been empirically and extensively verified in the work of *[Yu et al, 2020]*, in both supervised and self-supervised settings.
>
>
> **Q: Obtaining convolution without shift invariance.** ''Is it possible for the objective leads to convolution without explicit inducing shift invariance?''
>
> **A:** One may always choose the operators in each layer of a network to be convolutions to ensure either equivariance or invariance (as suggested in the work *[Cohen and Welling, 2016]* "Group Equivariant Convolutional Networks"). However, even so, the roles (and parameters) of the convolutions in each layer are not specified, nor the reason why such convolutions need to be composed as multiple layers.  What makes this work very different from previous approaches is that the layered convolutions (and nominal values of their parameters) are derived from a single objective function that depends only on the data. To our best knowledge, rate reduction is the first objective function that has been shown to have such surprising properties: not only its gradient shares the characteristics of linear and nonlinear operators in a layer of a deep network, but also the gradient becomes multi-channel convolutions when the data are invariant. As invariance is a special case of equivariance, we believe precise roles of convolutions for an equivariant task (such as detection) can also be derived from certain objective function. But such an objective function remains elusive.
>
> **Q: Relating to other optimization inspired network design.** ''I think the paper could also related to optimization inspired network design, e.g. Optimization Algorithm Inspired Deep Neural Network Structure Design, which delivers another perspective.''
>
> **A:** We thank the reviewer for pointing out related research. We will cite this paper and add a discussion in the final version.

---

### Official Review · AnonReviewer5 · 2020-11-09
**An interesting paper that may leave several outstanding questions.**

**Rating:** 4
**Confidence:** 3

**Review:**

In their submission the authors discuss an alternative learning rule to backpropagation called “maximal coding rate reduction” that was introduced recently in Yu et a. 2020. As far as I could tell, as a non-expert, maximizing the coding rate reduction objective encourages inputs from different classes to be maximally incoherent with respect to one another (spanning the largest possible volume) while inputs from the same class should be highly correlated.

In the present submission, the authors show that gradient ascent on the coding rate reduction naturally takes the form of a “neural network” with a particular residual network architecture that they term ReduNet. The authors argue that sending an input through the layers of ReduNet naturally leads to representations that (approximately) maximize the coding rate. The authors continue to show that when the representation is required to be group-equivariant with respect to shifts, a convolutional structure naturally emerges. This allows the authors to compute the update using layerwise fourier transformations. Finally, the authors show that on several example tasks ReduNet leads to outputs that have a large coding rate with no parameters learned via backpropagation.

There were several aspects of this submission that I liked a lot. I think the construction seems interesting and the rate reduction metric seems like a reasonable thing to optimize. I found the relationship of coding rate maximization to ReduNet to be quite clever and the experiments seemed to suggest that the approximations employed in the paper (e.g. the estimated membership in eqn 9) did not spoil the mapping.

Having said this, I don’t know that I agree with the authors’ interpretation of their results and I believe they might be somewhat overstated -- of course I am happy to be corrected if I am incorrect in my assessment.

1) My primary concern is the claim that the maximum rate reduction principle gives first-principles insight into convolutional networks. The authors show that a particular convolutional architecture, namely ReduNet, approximates projected gradient ascent on the MCR2 metric. However, it is not at all clear to me that standard convolutional networks or residual network architectures are well-described by ReduNet. In fact, it seems that the very particular structure of ReduNet has a number of features that are at odds with standard convolutional networks: nonlinearity only enters into ReduNet via the approximate membership (which enters via a softmax-like function) and outputs of each layer are projected onto the (n-1)-sphere.

2) Since the relationship between ReduNet and commonplace CNNs seems tenuous to me, it seems there is a significant burden on the authors to probe this question empirically. For example, do CNNs trained via backpropagation lead to architectures that resemble ReduNet? Unfortunately, as far as I could tell, the authors did not include experiments of this type.

3) If the authors are proposing ReduNet as an alternative to modern CNNs, it would be nice to see how ReduNet performs on some common tasks with well-established baselines (possibly with some fine-tuning step). However, as far as I could tell most of the experiments focused on verifying the properties of ReduNet. Note that I do not think one needs to achieve state-of-the-art performance here, but it would be nice to see how ReduNet fairs since this will probably affect its impact.

A few more minor points:

1) I may be missing something, but given the construction of ReduNet, I feel as though the emergence of a convolutional structure subject to translation invariance is not terribly surprising. Indeed I feel like this result has high overlap with Cohen and Welling [1]. This is not necessarily a criticism of the result per-se, but I think the phrasing could do a bit more to note this connection.

2) It was not obvious to me why Z had to be constrained to S^{n-1}. Can the authors provide some intuition here?

3) The authors note that non-linearity enters the network via the estimated membership. However, as far as I could tell, the estimated membership is only used for test points. Does this imply that the network is linear when the membership is known?

[1] Cohen, T. and Welling M. “Group Equivariant Convolutional Networks” JMLR 2016

Update: After discussion with the authors, I am inclined to lower my score. While I find the architecture proposed by the authors to be interesting, I do not think they have done enough to motivate the connection with neural networks. I find this especially troubling since the language used by the authors continues to imply that the connection is obvious. I would encourage the authors to look into the literature on scattering networks (e.g. https://arxiv.org/abs/1203.1513) for another approach to explaining networks from first principles that, I think, does a better job of making the connection to realistic neural network architectures.

---

> ### Author Response · Authors · 2020-11-13
> **Response to AnonReviewer5 [part 1]**
>
> Thank you for your feedback on our work.
>
> **Q: Insights from rate reduction as first principle for convolutional networks.** ''My primary concern is the claim that the maximum rate reduction principle gives first-principles insight into convolutional networks. ... outputs of each layer are projected onto the (n-1)-sphere.''
>
> **A:** First of all, our goal is not to explain all structures and components in the architectures of modern CNNs, at least not precisely the way they are. But there are undeniable resemblances. Furthermore, our analysis helps parse out at least three different roles of nonlinearities in a network: the membership, the normalization, and the sparsity-promoting in the early feature lifting stage. Furthermore, the convolutional structures are not by choice nor by design, they are derived as necessary from a clear objective function on the desired features. This is very different from all previous approaches that take convolutions for granted, as we will elaborate on more in answer to your next question.
>
> **Q: Relation with [Cohen and Welling].**
>
> **A:** As we have replied to AnonReviewer1 on a similar question: one may always choose the operators in each layer of a network to be convolutions to ensure either equivariance or invariance (as suggested in the work *[Cohen and Welling, 2016]* "Group Equivariant Convolutional Networks"). However, even so, the roles (and parameters) of the convolutions in each layer are not specified, nor the reason why such convolutions need to be composed as multiple layers. What makes this work very different from previous approaches is that the layered convolutions (and nominal values of their parameters) are derived from a single objective function that depends only on the data. To our best knowledge, the rate reduction is the first objective function that has been shown to have such surprising properties: not only its gradient shares the characteristics of linear and nonlinear operators in a layer of a deep network, but also the gradient becomes multi-channel convolutions when the data are invariant. We are not aware any other objective function share the same properties. This is indeed very surprising, at least to us.
>
> **Q: Why Z had to be constrained to $\mathbb{S}^{n-1}$.** ''It was not obvious to me why Z had to be constrained to $S^{n-1}$. Can the authors provide some intuition here?''
>
> **A:** The reason for performing normalization is to meaningfully compare the amounts of rate reduction between different representations (by different networks or at different layers), which is justified in the work of *[Yu et al, 2020]*. For example, consider a set of representation $Z = [z^1, ..., z^m]$, if we multiply every $z^i$ with a constant $C$ larger than 1 ($C \gg 1$), then $\widehat{Z} = [C\cdot z^1, ..., C\cdot z^m]$ would achieve higher rate reduction loss, (by applying the property of the $\log \det$ function.) i.e., $\Delta R(\widehat{Z}) > \Delta R(Z)$. However, these two sets of representations are equivalent up to scaling. Notice that there is no unique way to normalize as long as all representations compared adopt the same normalization scheme. Normalizing all the features onto the sphere is more for its simplicity. This view may also help explain why there are many different normalization schemes (batch norm, group norm, instance norm, etc.) and they all seem to work reasonably well.
>
> **Q: The network is linear when the membership is known?** ''The authors note that non-linearity enters the network via the estimated membership. However, as far as I could tell, the estimated membership is only used for test points. Does this imply that the network is linear when the membership is known?''
>
> **A:** This is not true, in order to construct the network layer by layer, we need to apply the non-linearity to approximate the membership of the training data as defined in Eq. (9). More specifically, the training labels, which can be represented as $\Pi^{j}$, are used for computing $C_{\ell}^{j}$, and in order to get the feature $z_{\ell+1}$, we need to apply non-linearity as defined in Eq. (9) and Eq. (10). On the other hand, all the $E_{\ell}$ and $C_{\ell}^{j}$ computed based on the training data are available during the test, we only need to perform forward propagation by using the non-linearity. If we do not apply non-linearity for constructing the networking during training, then the forward propagation will be inconsistent between training and test.

---

> > ### Comment · AnonReviewer5 · 2020-11-20
> > **Some additional comments.**
> >
> > Thanks for your detailed reply, it has definitely helped me to resolve some questions I have about your paper.
> >
> > However, I still think I may disagree with how you are positioning your contributions. I'm happy to continue discussing to see if you can help me understand your perspective better.
> >
> > You write that there are "undeniable resemblances" between ReduNet and neural networks. Can you be precise about what you take these undeniable resemblances to be? To me, it seems that the compositional nature of the architecture is similar and both architectures use affine transformations interleaved with _some_ nonlinearity along with a residual connection. However, the specific structure is quite different (where the affine transformations are located and the role / type of the nonlinearity). My point is that these differences seem significant enough to put a burden on you to show neural networks are actually implementing any of the ReduNet structure.
> >
> > Next you write, "Furthermore, our analysis helps parse out at least three different roles of nonlinearities in a network: the membership, the normalization, and the sparsity-promoting in the early feature lifting stage." However, to me it seems like I already need to have bought into the "undeniable resemblance" for this to be considered an impact of your work. As I mention above, if trained neural networks don't actually resemble ReduNets then why should I think that the nonlinearities in NNs actually fulfill the suggested roles?
> >
> > Finally, you mention that "Furthermore, the convolutional structures are not by choice nor by design, they are derived as necessary from a clear objective function on the desired features. This is very different from all previous approaches that take convolutions for granted..." and "To our best knowledge, the rate reduction is the first objective function that has been shown to have such surprising properties: not only its gradient shares the characteristics of linear and nonlinear operators in a layer of a deep network, but also the gradient becomes multi-channel convolutions when the data are invariant. We are not aware any other objective function share the same properties. This is indeed very surprising, at least to us."
> >
> > I agree that your constructive approach is different from the Cohen and Welling perspective, thanks for clarifying! However, I simultaneously find this to be a misrepresentation of the Cohen and Welling work. As far as I am aware, they do not take convolutions for granted. They note that convolutions naturally arise whenever neural network layers are constructed to be equivariant with respect to the translation operator $(g *  f(x) = f(g * x))$ for all $g$ which is not so dissimilar from your explicit augmentation of the data to include $g * x$ for all $g$.

---

> > > ### Author Response · Authors · 2020-11-20
> > > **regarding similarity to neural network and invariance**
> > >
> > > Thank you for your prompt and insightful reply.
> > >
> > > We agree there are still differences between ReduNet and specific neural networks that people use in practice. Nevertheless, we would like to point out that the architectural differences between ReduNet and networks such as ResNet and ResNeXt, etc. are considerably less than differences among many existing neural networks themselves (VGG, Inception, ResNet, DenseNet, U-Net, etc). We do agree that further studies are needed regarding what additional assumptions could help revise the design and explain more properties of the existing neural network. ReduNet is just the first step in this direction.
> > >
> > > We do not understand what you mean by "is quite different where the affine transformations are located". At each layer, all the linear operators (including the multi-convolutions) of ReduNet act on the input exactly the same way as all conventional neural networks. The output of the linear operators then goes through nonlinear selection or normalization operators. In fact, similar mechanisms have been adopted in Capsule network or mixture of Experts network as well.
> > >
> > > Typically existing neural networks do not delineate nor clarify the role(s) of nonlinear operators -- many simply use ReLU across. As a byproduct of our derivation, it is revealed that these nonlinear operators may play different roles or have different functions at different stages or places of the network. They may be subject to different choices.
> > >
> > > Regarding invariance/equivariance, all existing networks, including Cohen and Welling, arrive at the form of convolutions by imposing equivariance as the desired property on the operator of each layer. Our work is different: we justify the necessity of convolutions from the perspective of optimizing a clear objective directly. One may argue the results look similar (and actually they should). However, please note that from our derivation, not only do we arrive at the form of convolutions (from the objective) but also the values of the parameters of those convolutions. We believe this is a major difference in our work from Cohen and Welling etc. We will definitely make this point more clear in our paper. Your comments are truly helpful.

---

> ### Author Response · Authors · 2020-11-13
> **Response to AnonReviewer5 [part 2]**
>
> *[Continue with the previous response]*
>
> **Q:** For example, do CNNs trained via backpropagation lead to architectures that resemble ReduNet?''
>
> **A:** There have been efforts in the literature that try to show that structures such as convolutions may emerge from training a fully connected deep network *[Neyshabur, 2019]*. However, to our best knowledge, no work has come close to show what other internal (blocks of) structures a well-trained deep network may have acquired from the learning process. But as we have attempted to argue in this paper that, many popular architectures that people have found through years of selection do draw a strong resemblance to the ReduNet. Now, since the ReduNet is a white box, it would be interesting to find out in the future if similar substructures as the $C^j$ and $E$ do arise in other networks.
>
> **Q:** ''If the authors are proposing ReduNet as an alternative to modern CNNs, ... its impact.''
>
> **A:** We are not proposing ReduNet as an alternative to modern CNNs. Rather to the opposite, the constructive nature of ReduNet justifies the structures of modern CNNs to a large extent. Nevertheless, the clear principles behind ReduNet might suggest new ways of revising or improving current CNNs. Note that the scope and goal of this paper are only to present the basic framework and verify the correctness of the proposed concepts. We believe it is very promising to modify and improve the basic ReduNet so that it scales up to more practical and challenging tasks. We will explore these directions in our future work.
>
> *[Neyshabur, 2019] Towards Learning Convolutions from Scratch. Behnam Neyshabur. NeurIPS 2020.*

---

### Decision · Program_Chairs · 2021-01-07
**Final Decision**

**Decision:**

Reject

**Comment:**

This paper received borderline scores, which makes for a difficult recommendation. Unfortunately, two of the reviews were too short and thus were of limited use in forming a recommendation. That includes the high-scoring one, which did not adequately substantiate its score.

There is much to admire in this submission. Reviewers appreciated the originality of this research, linking rate reduction optimization to deep network architectures:
* R1: "The paper proposes a novel perspective"
* R4: "The novelty of the paper is in that formulation of the feature optimisation is baked-in into a deep architecture"
* R5: " I think the construction seems interesting and the rate reduction metric seems like a reasonable thing to optimize. I found the relationship of coding rate maximization to ReduNet to be quite clever"
* R3 (short): "The innovative method allows the inclusion of a new layer structure named ReduNet"

Reviewers also applauded the paper's clarity, including R4 who raised their score to 6 based on satisfying clarity revisions from the authors:
* R1: "The writing is good and easy to follow"
* R4 post-discussion: "Clarity is not an issues anymore - additional explanations provided by the authors and one more careful reading of the paper helped in understanding of all the aspects of the model"
* R2 (short): "The paper is well-structured."

However, there were some core questions around how well the main significance claims of the paper are supported. The most in-depth discussion on these topics is in the detailed thread with R5. In that thread there are many points discussed, but the two issues seem to be:
1. whether the connection between ReduNet and standard neural net architectures is sufficiently substantiated so as to constitute an explanation for behaviors of those standard architectures, like CNNs; and
2. whether the emergence of ReduNet's group invariance/equivariance is surprising or qualitatively new.

The first is much more central. On the first issue, R5 writes in summary:
"Fundamentally I think the authors propose a hypothesis: that ReduNets explain DL models. However, the authors do not take meaningful steps towards validating this hypothesis. [...] I would contrast this with, for example, the scattering networks paper (https://arxiv.org/abs/1203.1513) which did an exceptional job of arguing for an ab initio explanation of convolutional networks."

I find R5's perspective on this point to be compelling, in that the paper currently doesn't do enough to justify these main claims, either through drawing precise nontrivial mathematical connections or through experimental validation. (The thread has a much more detailed and nuanced discussion.)

The second issue is not quite as central to the significance of the paper, but it was noted by multiple reviewers:
* R5: "I may be missing something, but given the construction of ReduNet, I feel as though the emergence of a convolutional structure subject to translation invariance is not terribly surprising."
* R4: "Finally, I am not sure if the result of obtaining a convnet architecture in ReduNet when translation invariance constraint is added the embedding is all that surprising."
* R4 post-discussion: "Reading the exchange between the authors and R5 I am still not fully convinced that translation invariance property is all that surprising, but for me that's not a reason to reject."

At the least, the paper as written hasn't yet convinced some readers (myself included) on these claims.

As I mentioned at the start, this paper is borderline, but because I am largely aligned with R5's perspectives, I think this paper does not quite pass the bar for acceptance. I recommend a rejection, but I look forward to seeing a strengthened version of this work in the future. I hope the feedback here has been useful to bringing about that stronger version.